# Evolution of biogeochemical Properties Inside Poleward Undercurrent Eddies in the Southeast Pacific Ocean

Lenna O. Ortiz-Castillo[1,2,3], Oscar Pizarro[3,4,9], Marcela Cornejo-D'Ottone[5] and Boris Dewitte[6,7,8]

[1]Centro de Investigaciones en Ecosistemas de la Patagonia (CIEP), Coyhaique, Chile,
[2]Programa de Postgrado en Oceanografía, Departamento de Oceanografía, Facultad de Ciencias Naturales y Oceanográficas, Universidad de Concepción, Chile,
[3]Millennium Institute of Oceanography, Universidad de Concepción, Chile
[4]Department of Geophysics, University of Concepcion, Chile,
[5]Escuela de Ciencias del Mar and Núcleo Milenio para el estudio de la Desoxigenación del Océano Pacífico Sur oriental (DEOXS), Pontificia Universidad Católica de Valparaíso, Valparaíso, Chile,
[6]Centro de Estudios Avanzados en Zonas Áridas (CEAZA), Coquimbo, Chile
[7]Departamento de Biología Marina, Facultad de Ciencias del Mar, Universidad Católica del Norte, Coquimbo, Chile,
[8]CECI, Université de Toulouse III, CERFACS/CNRS, Toulouse, France
[9]Centro de Instrumentación Oceanográfica, Universidad de Concepción, Chile

*Correspondence to*: Lenna O. Ortiz-Castillo (lenna.ortiz@ciep.cl) and Oscar Pizarro (oscar.pizarro@imo-chile.cl)

**Abstract**. Oceanic eddies are ubiquitous features of the circulation thought to be involved in transporting water mass properties over long distances from their source region. Among these is a particular type with a core within the thermocline with little surface expression. Despite their significance, their role in the ocean circulation remains largely undocumented from observations. This study characterizes the variations in internal biogeochemistry, disparities with external properties, and processes influencing the dissolved oxygen budget of Poleward undercurrent eddies (Puddies) during their transit to oceanic waters. Employing a high-resolution coupled simulation of the Southeast Pacific, we document biogeochemical properties and processes associated with the nitrogen cycle inside Puddies and contrast them with those of the surrounding environment. Our findings reveal that Puddies capture a biogeochemical signal contingent upon their formation location along the coast, particularly associated with the core of the Peru-Chile Undercurrent at the core of the Oxygen Minimum Zone (OMZ). While permeability at the periphery facilitates exchange with external waters, thereby modulating the original properties, the core signal retains negative oxygen ($O_2$) anomalies and positive anomalies of other biogeochemical tracers. These anomalous conditions result in tracer values exceeding the 90th percentile of their distribution in the open ocean, in contrast to the formation zones, where anomalies only surpass the 50th percentile. This indicates that Puddies may play a role in modulating the average properties of the open ocean. Suboxic ($O_2 < 20$ μM) cores are prevalent near the coast but decrease in abundance with distance from shore, giving way to a predominance of hypoxic (20 μM $< O_2 <$ 45 μM) cores (predominating at 60% in the open sea), suggesting core ventilation during transit. The principal mechanism governing $O_2$ input into, or output from the eddy core entails lateral and vertical advection, with vertical mixing supplying $O_2$ to a lesser extent. Biological activity consumes $O_2$ inside Puddies for around 6 to 12 months, especially intensely for the first 100 days, thus facilitating the persistence of low $O_2$ conditions and extending the lifetime of biogeochemical anomalies within the core (up to 800 km offshore). The ammonium and nitrite deplete earlier

in the eddy core with a decay rate greater than the nitrate and nitrous oxide, while these are accumulating in open sea (up to 16% and 100% higher than mean state, respectively). Our results suggest that southern regions of the southeast Pacific OMZ undergo greater deoxygenation and nutrient enrichment due to Puddies compared to northern regions. However, the combination of various physical conditions can generate zones with more pronounced changes in the nitrite (subsurface water masses due to interactions with the Puddies, such as at 30ºS. The maximum contribution of $NO_2^-$) takes place in particular along this latitude, with a 460% increase compared to the mean state, near the coastal zone.

In summary, Puddies formed along the Chilean coast capture distinct biogeochemical "signatures" depending on where they form. In the north, minimal ventilation fosters suboxic conditions and denitrification –leading to deficits of nitrate ($NO_3^-$) and nitrous oxide ($N_2O$) but high $NO_2^-$ and ammonium ($NH_4^+$)– whereas central and southern subregions show increased $NO_3^-$ and higher $N_2O$. Moreover, cross-shore exchange between Equatorial Subsurface Water and Subantarctic Water further amplifies this variability, giving rise to eddies with diverse nutrient and oxygen properties as they move offshore.

## 1 Introduction

Oxygen plays a fundamental role for life in the ocean, and numerous processes regulate its concentration in the water column. In subsurface waters (100-800 m depth), oxygen concentrations ($O_2$) decrease significantly due to the decomposition of organic matter and limited ventilation, leading to the formation of oxygen minimum zones (OMZ) in eastern boundary upwelling systems (EBUS; Wyrtki, 1962; Helly & Levin, 2004; Karstensen et al., 2008; Paulmier & Ruiz-Pino, 2009; Stramma et al., 2010). Under these conditions of low oxygen in the subsurface (i.e. dissolved $O_2$ less than 20 μM), heterotrophic metabolic processes are important, dominated by activity of bacteria and archaea, resulting in significant shifts in biogeochemical cycles (Lam et al., 2009; Paulmier & Ruiz-Pino, 2009; Wright et al., 2012).

The nitrogen cycle in the oceans involves various chemical species with different oxidation states. Outside the OMZ, where conditions are oxygenated, dinitrogen ($N_2$) is transformed into ammonium ($NH_4^+$), nitrite ($NO_2^-$), and nitrate ($NO_3^-$) through nitrification, with nitrous oxide ($N_2O$) produced as a byproduct. However, within the OMZ, where $O_2$ is depleted, nitrate becomes the primary oxidant, triggering denitrification. In this process, nitrate is reduced to gaseous forms ($N_2$ and $N_2O$), which can then be released into the atmosphere. This process has implications for primary production, carbon sequestration, and the release of $N_2O$ into the atmosphere, a potent greenhouse gas (Goreau et al., 1980; Mantoura et al., 1993; Sarmiento & Gruber, 2006; Lam et al., 2009; Paulmier & Ruiz-Pino, 2009; Wright et al., 2012).

The southeast Pacific Ocean is the site of one of the most extensive and shallow OMZs (Paulmier & Ruiz-Pino, 2009), where even anoxic conditions can be observed (Ulloa et al., 2012). Several authors have determined the vertical and zonal (offshore) extent of the OMZ, which exhibits significant seasonal variability, modulated both meridionally by subsurface currents towards the pole, and zonally by mesoscale processes (jets, eddies, fronts, filaments, etc.; Bettencourt et al., 2015; Chaigneau et al., 2011; Grados et al., 2016; Hormazabal et al., 2013; Morales et al., 2012; Stramma et al., 2013; Vergara et al., 2016; Pizarro-Koch

et al., 2019). These processes result in changes in water mass properties, and together contribute up to a 25%
reduction in $O_2$ volume during spring (Pizarro-Koch et al., 2019).
Furthermore, future projections suggest the expansion of these $O_2$ depleted zones through global warming
(Matear & Hirst, 2003; Stramma et al., 2010; Oschlies et al., 2018) although uncertainties remain (Almendra
et al., 2024). The increase in sea surface temperature will affect $O_2$ solubility, and enhanced water column
stratification will impact a range of biological processes that influence $O_2$ concentrations (Couespel et al.,
2019; Keeling et al., 2010; Matear & Hirst, 2003; Oschlies et al., 2018; Schmidtko et al., 2017). In addition,
various mechanisms can potentially modify ventilation processes, leading to changes in subsurface water
properties. While in current generation climate models, the changes in mean circulation either in the tropics
or the high-latitudes, mediated by diapycnal mixing, are invoked to explain the ventilation process (Pitcher et
al., 2021), regional modeling studies highlight the importance of mesoscale dynamics, such as mesoscale
eddies and zonal jets, in expanding and ventilating oceanic zones with oxygen deficits (Bettencourt et al.,
2015; Auger et al., 2021; Calil, 2023). The regional models also provide a more realistic representation of the
OMZs than the climate models that do not resolve sub to mesoscale dynamics. This is why understanding the
role of mesoscale dynamics on the $O_2$ and carbon cycles in EBUS has been a key focus of research in recent
years.
'**P**oleward **u**ndercurrent e**ddies**' (Puddies) are types of subsurface or intrathermocline eddies characterized by
coherent anticyclonic lenticular shaped vortices with cores located within the pycnocline and relatively
homogenous interior waters (Dugan et al., 1982; McWilliams, 1985; Kostianoy and Belkin, 1989). Puddies
originate in the EBUS (Frenger et al., 2018) due to the interaction of the poleward-flowing current with the
continental slope, generating submesoscale instabilities with anticyclonic vorticity, subsequently forming
these characteristic mesoscale structures (Hormazabal et al., 2013; Combes et al., 2015, Molemaker et al.,
2015; Thomsen et al., 2016; Contreras et al., 2019). These eddies represent 30-55% of the anticyclone eddies
originating in the EBUS (Pegliasco et al., 2015; Combes et al., 2015) with cores that are warmer, saltier, $O_2$-
depleted, and nutrient-enriched relative to surrounding waters (Collins et al., 2013; Hormazabal et al., 2013;
Morales et al., 2012; Johnson and McTaggart, 2010). Therefore, Puddies can play a crucial role in
transporting these water properties hundreds or thousands of kilometers offshore to subtropical gyres,
potentially contributing to the expansion of the OMZ beyond the coastal region (Frenger et al., 2018).
Observations of low $O_2$ events in open ocean regions provide support for this idea (Lukas and Santiago-
Mandujano, 2001; Johnson and McTaggart, 2010; Cornejo-D'Ottone, 2016; Schütte et al., 2016b; Stramma et
al., 2013, 2014; Karstensen et al., 2015), with further evidence of biogeochemical processes typically
observed only in the OMZ, such as $N_2O$ production (Cornejo-D'Otonne et al., 2016; Arévalo-Martínez et al.,
2016; Grundle et al., 2017) and nitrogen loss through denitrification (Altabet et al., 2012; Löscher et al.,
2015). Additionally, the observed $O_2$ utilization rates within the eddy cores range from 0.29 to 44 nmol $O_2$ L$^-$
$^1$ d$^{-1}$, which is up to 3 to 5 times higher than in the surrounding waters (Cornejo-D'Ottone et al., 2016;
Karstensen et al., 2015). There is evidence of harboring microbial communities and metabolisms associated
with low-oxygen environments that persist even when the eddies enter highly oxygenated waters, a
phenomenon known as the "stewpot effect" (Löscher et al., 2015; Frenger et al., 2018).

The Southeast Pacific (SEP) is characterized by extensive subsurface mesoscale eddy activity with radii ranging from ~25 to ~50 km and cores of ~500 m of vertical extent (Chaigneau et al., 2009; Hormazabal et al., 2013; Combes et al., 2015, Frenger et al., 2018). They transport a total volume of approximately 1 Sv (1 Sv = ~1x10$^6$ m$^3$ s$^{-1}$) westward with an average velocity of ~2 km d$^{-1}$ (Hormazabal et al., 2013). The cores of these eddies exhibit homogenous salinity profiles (>34.5) and low O$_2$ concentrations (< 1.0 mL L$^{-1}$), linking them to Equatorial Subsurface Water (ESSW) transported poleward by the Peru-Chile Undercurrent (PCUC; Hormazabal, 2004; Colas et al., 2012; Hormazabal et al., 2013). Generally, the low values of O$_2$ in subsurface eddies is related to higher concentrations of nitrate, phosphate, and silicate (Czeschel et al., 2015). However, under suboxic conditions (O$_2$ < 20 μM), the prevailing anaerobic metabolism is denitrification, where nitrate is utilized as an electron acceptor, leading to increased production of NO$_2^-$ and N$_2$O (Goreau et al., 1980; Mantoura et al., 1993; Lam et al., 2009; Wright et al., 2012). Within these eddies, various biogeochemical processes coexist that are highly sensitive to O$_2$ variations, while physical processes modulate biogeochemical patterns through vertical and lateral mixing, submesoscale stirring, or mass exchange with water masses from different origins through turbulent advection (José et al., 2017; Kartensen et al., 2017; Loveccio et al., 2022).

This complexity of processes involved during the life cycle of a Puddy, along with the lack of continuous in situ measurements, limits our understanding of nutrient recycling throughout their lifetime, and the balance between processes controlling the rate of change of O$_2$ and nutrients. In the present study, we aim to characterize the internal biogeochemistry of eddies formed under various low-oxygen conditions in the SEP. We analyze the physical and biogeochemical factors influencing the variability in the lifespan of elements trapped within the Puddies, with a particular focus on those related to the nitrogen cycle, as they travel across the OMZ towards better-ventilated oceanic waters. Specifically, we document the evolution of water mass properties and processes inside Puddies with contrasting initial O$_2$ concentrations (suboxic versus hypoxic) to evaluate their role in maintaining the OMZ. Additionally, we aim to address knowledge gaps in the biogeochemical dynamics of this type of eddy, stemming from the lack of observational data, particularly in the SEP. To quantify the changes Puddies undergo due to their complex dynamics, we use a regional coupled biogeochemical model simulation.

Our approach involves a robust statistical analysis of contrasting water mass properties inside and outside the Puddies. First, we characterize the initial biogeochemical signal from their formation to assess how, and to what extent, the biogeochemical properties of Puddy' cores vary across different regions of the SEP. Second, we quantify how these properties influence the subsurface waters through which Puddies travel, evaluating the balance of processes involved.

The structure of this study is as follows: Section 2 details the model and methods used for the identification and characterization of Puddies. Section 3 describes the biogeochemical characterization inside and outside the Puddies, changes in the O$_2$ budget, and variations in biogeochemical properties from the coastal to oceanic zones. Section 4 discusses the results, and Section 5 presents the main conclusions and future projections.

## 2 Methods

### 2.1 Regional biogeochemical coupled model

We used a high-resolution, coupled physical-biogeochemical model simulation of the SEP that considers the main processes involved in the transformation of the water masses relevant to the OMZ variability and the dynamics of the Peru-Chile Undercurrent (PCUC). This current plays a significant role in generating Puddies and in the southward extension of the OMZ in the Peru-Chile EBUS.

The physical dynamics was simulated using the Regional Ocean Modelling System (ROMS), a regional ocean circulation model that solves the primitive equations with free surface and sigma coordinates (Shchepetkin and McWilliams 2005; 2009). ROMS was coupled with the biogeochemical model BioEBUS, specifically developed for Eastern Boundary Upwelling Systems (EBUS) and based on the nitrogen cycle using $N_2P_2Z_2D_2$ model formulation (Koné et al., 2005; Gutknecht et al., 2013a). We adopted the same configuration as specified in several other studies in the region (Dewitte et al., 2012; Vergara et al., 2016; Pizarro-Koch et al., 2019). The model has a spatial resolution of 1/12°, 37 vertical levels. 3-day mean outputs are used, which is suitable for resolving mesoscale features. The overall domain covers the latitudinal range of 12°N to 40°S from the coast to 95°W, although the present study focuses on latitudes off the coast of Chile between 20°S and 40°S (Figure 1a). The simulated period was from 2000 to 2008.

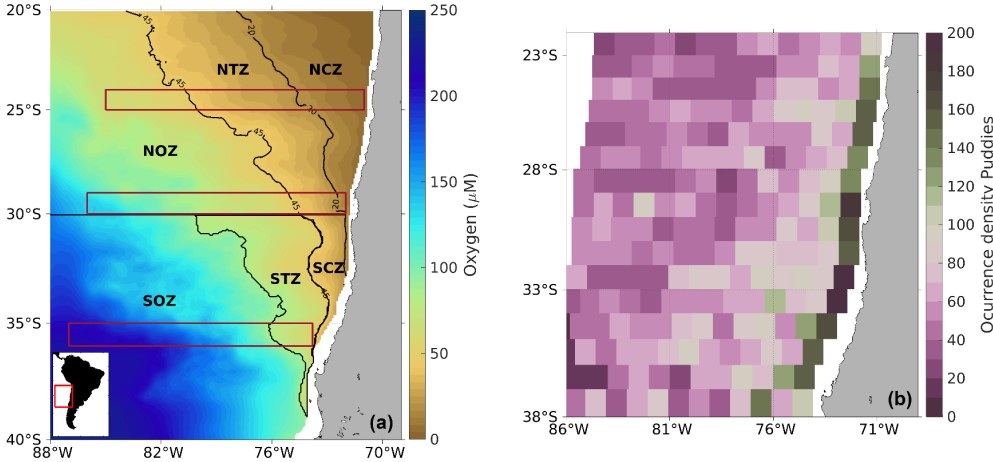

**Figure 1. (a) Modeled dissolved O$_2$ climatology off the Chilean coast, and (b) spatial distribution of the occurrence density of Puddies identified over the nine-year period. Oxygen was considered as being in the isopycnal layer 26.6 kg m$^{-3}$ representative of the OMZ core (defined with S$_{core}$ layer; see Section 2.2). The study area covers 20 - 40ºS and from the coast to 88ºW. The contours used to subdivide regions are based on biogeochemical differences among water masses (see details in Section 2.2). Northern Coastal Zone (NCZ) is delimited by the suboxic zone limit (O$_2$ = 20 µM); Northern Transition Zone (NTZ) and Southern Coastal Zone (SCZ) separate to 30ºS and the traditional OMZ limit (O$_2$ = 1mL L$^{-1}$ or ~ 45 µM), Northern Oceanic Zone (NOZ) with O$_2$ > 45 µM to northern 30ºS; Southern Transition Zone (STZ) delimited with oxygen contours of 45 µM < O$_2$ < 90 µM and Southern Oceanic Zone (SOZ) with 90 µM < O$_2$, both southern 30ºS. Red boxes enclosed the zonal bands taken for made the**

**average compounds by Puddies to 25ºS (from 71ºW to 85ºW), 30ºS (72ºW to 86ºW) and 36ºS (74ºW to 87ºW; see**
**Sections 2.6.2 and 3.4). (Right box) Occurrence density (colormap) is quantified by the total number of Puddies**
**identified in each 1°x1° area for each snapshot (Δt = 3 days), with the possibility that the same eddy may be**
**counted more than once if remaining in the same area. Numbers refer to the total occurrence density by region**
**(see details in Section 2.5).**

The model uses atmospheric momentum forcing data from NCEP-NCAR statistically downscaled based on
the QuickSCAT satellite data (see Goubanova et al. (2011) for details). Latent heat flux and other variables –
for estimating other air-sea fluxes– such as air temperature and humidity are provided by monthly
climatology with a resolution of 1° x 1° from COADS (da Silva et al., 1994). The boundary conditions for
temperature, salinity, and horizontal velocity were provided by the SODA 1.4.2 reanalysis (Smith et al.,
1992). The BioEBUS model consists of 12 compartments interacting through advection-diffusion equations
and source-minus-sink (SMS) processes. The considered components include inorganic dissolved nutrients
($NO_3^-$, $NO_2^-$, and $NH_4^+$), large and small phytoplankton ("small" representing nanophytoplankton; mainly
small flagellates between 2 and 20 µm and "large" representing microphytoplankton; mainly diatoms
between 20 and 200 µm), large and small zooplankton ("small" representing microzooplankton; mainly
heterotrophic ciliates between 20 and 200 μm, and "large" mesozooplankton; mainly copepods between 200
μm and 2 mm), and detritus (small and large). Dissolved organic nitrogen (DON) was considered following
the formulation of Dadou et al. (2001, 2004) and Huret et al. (2005), $O_2$ including its ocean-atmosphere
interaction according to Peña et al. (2010) and Yakushev et al. (2007), and the production of $N_2O$ using the
parameterization of Suntharalingam et al. (2000, 2012). The boundary and initial conditions for the BioEBUS
model were obtained from the CARS2006 climatology (CSIRO Atlas of Regional Seas) for $O_2$ and $NO_3^-$,
with constant vertical profiles adopted for $NH_4^+$, $NO_2^-$, and dissolved organic nitrogen (DON) (based on
Koné et al. (2005)). The estimated phytoplankton biomass comes from the model. Detailed information on
simulation and validation of the physical model (ROMS) is given by Dewitte et al. (2012) and Vergara et al.
(2017). The parameter configuration of BioEBUS is the same as that used in Montes et al. (2014).
The time rate of change of the concentration of each component is governed by the advection-diffusion
equation (see Gutknecht et al., 2013a). For instance, the $O_2$ balance is given by
$$\frac{\partial O_2}{\partial t} = -\nabla \cdot (\vec{u} O_2) + K_H \nabla^2 O_2 + \frac{\partial}{\partial z}\left(K_z \frac{\partial O_2}{\partial z}\right) + SMS(O_2) \tag{1}$$

where $\vec{u} = (u, v, w)$ represents the fluid velocity, with components $u$ for zonal, $v$ for meridional, and $w$ for
vertical. The first term on the right-hand side represents the advection ($ADV = -\nabla \cdot (\vec{u} O_2)$), which is a
scalar, but can also be decomposed in the sum of zonal ($XADV = -u \frac{\partial O_2}{\partial x}$), meridional ($YADV = -v \frac{\partial O_2}{\partial y}$),
and vertical ($VADV = -w \frac{\partial O_2}{\partial z}$) components. The second and third terms correspond to horizontal (HMIX)
and vertical (VMIX) diffusion, where $K_H$ is the horizontal eddy diffusion coefficient (set to 100 m$^2$ s$^{-1}$ in this
version of the model), and $K_z$ is the turbulent diffusion coefficient calculated using the K-profile
parameterization mixing scheme (Large et al., 1994). The last term $SMS(O_2)$ represents the effect of sources
and sinks associated with biogeochemical processes. For $O_2$, the source process is primary production, and
sink processes include remineralization, nitrification, and zooplankton excretion (Peña et al., 2010).

**2.2 Characterization of the study area**

The study area extends from 20º to 40ºS and from the Chilean coast to 88ºW. In this region the OMZ core
($O_2 < 45\ \mu M$ or $\sim 1$ mL/L) is centered at a density surface of $\sigma_\theta = 26.6$ kg m$^{-3}$ ($S_{core}$) following as reference
the core of ESSW near coastal zones, which is similar to Pizarro-Koch et al., (2019; see Section 2.2). $S_{core}$
depth varies, being shallow in the coastal area and deeper in the oceanic region (Table 1). Near the slope, a
deepening in $S_{core}$ is observed north of 30°S, where the slope is narrower whereas south of 30°S, it widens.
All variables were interpolated from the original sigma vertical coordinate to depth at 5-meter intervals from
800 m to the surface. In the model, there are 28 vertical levels spanning ~1000 m depth to the surface and 9
vertical levels for depths >1000 m (Figure S1). Regions were selected for averaging tracers and rates, which
was based on the characteristic water masses in the region, resulting in domains with irregular shapes (see
Figure 1a). The water masses that were considered include primarily the Subantarctic Water (SAAW; also
called East-southern Pacific Intermediate Water (ESPIW), characterized here by 11.5°C, 33.8) south of 30°S,
where the OMZ is predominantly hypoxic ($O_2 < 45\ \mu M$; as described by Naqvi et al., 2010, and Pizarro-Koch
et al., 2019) and significantly narrower. The biogeochemistry of SAAW differs notably from that of
Subtropical Water (STW; 20°C, 35.2) north of 30°S, which features a zonally broader OMZ characterized by
suboxic conditions ($O_2 < 20\ \mu M$, Figure 2; following Wright et al., 2012), particularly in oxygen,
ammonium, and nitrite.
The separation of coastal, transitional, and oceanic zones considered the dominance of Equatorial Subsurface
Water (ESSW; 12.5°C, 34.9) in the subsurface waters and its increasing oxygenation farther from the coast
(Silva et al., 2009). Given the importance of ESSW in defining the study areas and as the primary water
source enclosed in the Puddies, the isopycnal surface associated with the ESSW and OMZ core was used as a
reference. Specifically, we focused on the isopycnal with a density of 26.6 kg m$^{-3}$, corresponding to the
reference OMZ core in this model (Pizarro-Koch et al., 2019), which was used in part of our statistical
analysis.
We thus came up with 6 subregions: Northern Coastal Zone (NCZ; suboxic conditions); Northern Transition
Zone (NTZ; hypoxic conditions); Northern Oceanic Zone (NOZ; with $45\ \mu M < O_2$); Southern Oceanic Zone
(SOZ; with $90\ \mu M < O_2$); Southern Transition Zone (STZ with $45\ \mu M < O_2 < 90\ \mu M$) and Southern Coastal
Zone (SCZ; hypoxic conditions, Figure 1a).

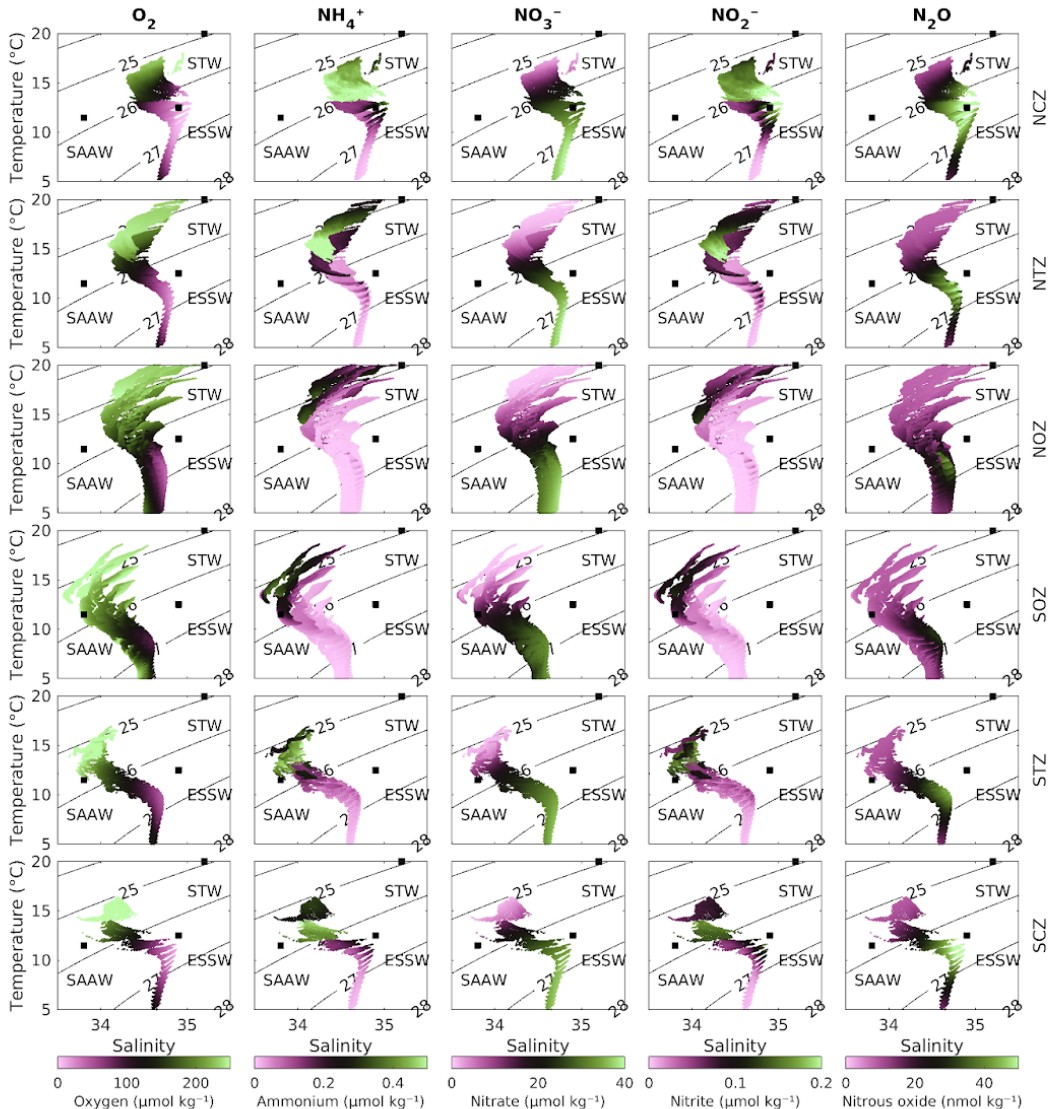


**Figure 2. Diagram showing the averages of conservative temperature, absolute salinity, and biogeochemical tracers (in color) for the six study zones (see Figure 1a). The biogeochemical tracers include oxygen (column 1), ammonium (column 2), nitrate (column 3), nitrite (column 4), and nitrous oxide (column 5). From top to bottom the region's names are : Northern Coastal Zone (NCZ); Northern Transition Zone (NTZ); Northern Oceanic Zone (NOZ); Southern Oceanic Zone (SOZ); Southern Transition Zone (STZ) and Southern Coastal Zone (SCZ). The zones' selection is based on the characteristic water masses in the region, primarily the SAAW (Subantarctic Water) south of 30°S, whose biogeochemistry differs significantly from the STW (Subtropical Water) north of 30°S, particularly in oxygen, ammonium, and nitrite. The separation of the coastal, transitional, and oceanic zones considered the dominance of the ESSW in the subsurface waters and its oxygenation away from the coast.**

Within the first ~100 km off the coast, the formation zone for Puddies was identified, where a large number of surface and subsurface eddies are typically generated (e.g., Chaigneau et al., 2009) as shown by the occurrence density Puddies (Figure 1b green strip). We considered the total number of Puddies's profiles with a grid box of approximately 1° by 1° in order to characterize the biogeochemical properties of the source water that the Puddies eventually enclose upon formation (see Section 2.6.1).

**2.3 Definition of "mean state" and mesoscale contribution**
To estimate the physical and biogeochemical perturbations from the mean field associated with the eddies,
we use a Reynolds decomposition, which for the field of $NO_3^-$ concentration is written as follows in Eq. (2):
$$NO_3(x,y,z,t) = <NO_3>_{9-year} + NO_3{'}(x,y,z,t)$$    (2)
where $<NO_3>_{9-year}(x,y,z)$ is the "mean state" of $NO_3^-$ calculated as an overall mean across 9 years
simulated between the period 2000-2008. Fluctuations of this "mean state" thus consider the large
intraseasonal and seasonal variability and are denoted as $NO_3{'}(x,y,z,t)$. Similarly, for the other variables, $<$
$S>_{9-year}, <NO_2>_{9-year}$, etc., denoted the "mean state", whereas $S'$, $NO_2{'}$ etc., correspond to anomalies,
which includes eddy fluctuations and changes associated with annual and interannual variability. The
averages anomalies were denoted as $<S'>, <NO_2{'}>$, etc., This decomposition method was used for all
variables analyzed in each selected subregion (Figure 1a). To evaluate the impacts of the Puddies on the
various fields, we used an algorithm to identify subsurface eddies (See details in Section 2.5) and then
compared the perturbed fields inside the Puddies with the total field. This latter procedure is further
explained in section 2.6.
**2.4 Calculation of AOU, $\Delta NO_3^-$ and $\Delta N_2O$**
The apparent oxygen utilization (AOU), $NO_3^-$ production ($\Delta NO_3^-$), and $N_2O$ production ($\Delta N_2O$) provide an
estimate of how much has been produced/consumed by biological processes since the water mass was formed
in terms of oxygen, nitrate, and nitrous oxide respectively. These estimates are associated with the time the
water mass has spent without coming into contact with the ocean surface or being ventilated.
The AOU calculation was derived from the García & Gordon (1992) algorithm, based on the $O_2$ saturation
concentration at any temperature and salinity, ):
$$AOU = [O_2]_{sat} - [O_2]$$    (3)
where $[O_2]_{sat}$ is the oxygen saturation and was calculated based on TEOS-10 (using the Gibbs-SeaWater
oceanographic toolbox) (https://www.teos-10.org/software.htm), for $[O_2]$ we used the modeled oxygen. The
$N_2O$ production ($\Delta N_2O$) was calculated using the Gruber & Sarmiento (2002)'s methodology and the
following relationship, Eq. (4):
$$\Delta N_2O = [N_2O] - [N_2O]_{sat}$$    (4)
where $[N_2O]_{sat}$ is the nitrous oxide saturation calculated from to modeled temperature and salinity, for $[N_2O]$
we used the modeled nitrous oxide. $NO_3^-$ production ($\Delta NO_3^-$) is defined as follows,
$$\Delta NO_3^- = [NO_3^-] - [NO_3^-]_{preformed}$$    (5)
where [NO$_3^-$] is the modeled nitrate. The value of [NO$_3^-$]$_{preformed}$ in subsurface waters considered for the
above calculation was that of Equatorial Subsurface Water (ESSW), except in SOZ where the value for
Antarctic Subsurface Water (SAAW) was taken from Llanillo et al. (2012). The assessment of the modeled
O$_2$, NO$_3^-$ and surface NO$_3^-$ is provided in Appendix.

### 2.5 Puddy Identification

For the identification of Puddies, the algorithm proposed by Faghmous et al. (2015) was adapted to deal with
subsurface eddies that have a weak dynamical signature at the surface of the ocean. The original algorithm is
based on the presence of local extreme values of sea level anomalies (SLA) (minimum in the case of cyclonic
eddies and maximum in the case of anticyclonic eddies, considering a neighborhood defined a priori around
it). Because the SLA signal from subsurface eddies may be rather weak or absent, the present study used
anomalies in the layer thickness ($\delta h$) defined as the difference between the depths of the density surfaces
S$_{upper}$ = 26.0 kg m$^{-3}$ and S$_{lower}$ = 26.9 kg m$^{-3}$. Thus, positive anomalies ($\delta h > 0$) indicate the presence of
subsurface anticyclonic eddies due to their convex shape (for our case, Puddies). When $\delta h$ is at its maximum
($\delta h_{max}$), the largest closed contour around the geographical location of $\delta h_{max}$ is considered as the edge of the
eddy, as $\delta h_{max}$ is associated with the center of the eddy and the points contained within the eddy edge are the
body of the eddy (Figure S2a; Faghmous et al., 2015). Starting from $\delta h_{max}$, a gradual decrease of 0.1 m was
used to establish the size and amplitude of the eddy. Only eddies that reached a minimum horizontal area of
30 grid points (A$_{min}$~1.95x10$^9$ m², equivalent to a radius of ~ 25 km) were considered, as eddies below this
threshold are not well identified (reducing the total of selected Puddies by 30%). Additionally, only eddies
with a radius not exceeding 150 km (~32 pixels in diameter) were included, as larger eddies are also not well
detected, with at most three closed-contour structures per snapshot being discarded. This also helps
distinguish eddies in coastal regions from other processes, such as coastal trapped waves or upwelling events
which occur on larger spatial scales.
Eddy identification was performed using 3-day averages across the entire period. Subsequently, all $\delta h_{max}$
positions were classified by subregion and in 1°x1° cells for their enumeration (Figure 1b) and for
characterizing the average properties of these Puddies along the coast (see Section 2.6.1; statistical analyses
are shown in section 3).

### 2.6 Compound Formation

### 2.6.1 Average Puddies Profiles

To understand the typical conditions within Puddies identified in the formation zone and each subregion,
average profiles were constructed as follows: i) all eddy centers (i.e., $\delta h_{max}$ positions) were classified in
1°x1° grid cells along the coast and also within the regions defined in Figure 1a, ii) for each eddy center,
vertical profiles were extracted between the density surfaces S$_{upper}$ and S$_{lower}$ for all variables, in each
corresponding region, iii) then, these profiles were time-averaged to obtain typical profiles for each variable.
For the different regions, the values associated with the $S_{core}$ are shown (Section 3.2) to emphasize changes in
the ESSW core. The analysis of these results is presented in sections 3.1 and 3.2.

**2.6.2 Average zonal compound**

Based on the identification of eddies over a 9-year period, different latitude bands (25°S, 30°S, and 36°S,
Figure 1a) were selected to create a composite constructed from 3D Puddies, considering all the $\delta h_{max}$ values
identified within 1°x1° areas. For these Puddies, we estimated characteristics within the entire eddy volume
as follows: i) for each $\delta h_{max}$ (Puddy center), a mask of the entire volume was created; ii) the area mask was
calculated to identify the edge using the Faghmous et al. (2015)'s algorithm, which was found to be most
accurate approach (Figure S3; see sensitivity analysis in the Supplementary Material); iii) this mask was
applied across all depths (at 5-meter intervals), creating an irregular cylinder between $S_{upper}$ and $S_{lower}$; iv)
from the total volume enclosed by the cylinder, an average vertical profile was obtained at each time step for
each variable (see an example of vertical Puddy characterization in Figures S2b-e).
A total of 862 Puddies ($\delta h_{max}$ values) were accounted for in the 25°S zonal band, 810 in the 30°S band, and
658 in the 36°S band. Additionally, for each latitude, biogeochemical variables averaged over the 9-year
period, representing the mean state (defined in Section 2.3), were used to calculate an average profile
associated with a 1°x1° area. Anomalies were then calculated as the difference between these two profiles.
The results of this statistical analysis and the subsurface modifications caused by Puddies are presented in
Section 3.4.

**2.7 Calculation of percentiles**

To assess the significance of the internal contribution of Puddies relative to the mean state (defined in
Section 2.3), we employed a bootstrap method, which involved randomly sampling more than 200 positions
in each region and deriving diagnostics. A total of 92 random time steps were selected, and for each time
step, 1,000 latitudes and 1,000 longitudes were generated within the domain. These positions were then
filtered for each region. The total number of random positions found per region was as follows: 1,246 in
NCZ, 1,474 in NTZ, 3,509 in NOZ, 6,808 in SOZ, 521 in STZ, and 235 in SCZ (Table S2).
For each obtained position, biogeochemical properties were extracted at the $S_{core}$. This approach generated an
ensemble of values, allowing us to infer the 50th, 75th, and 90th percentiles (P50, P75, P90) used to provide
an estimate of significance levels.

**3 Results**

**3.1 Contrasting biogeochemical characteristics inside and outside the Puddies in the formation zone**

The region within the first ~100 km off the coast was considered a formation zone for Puddies where a large
number of surface and subsurface eddies are typically generated (e.g., Chaigneau et al., 2009; Figure 1b,
green strip). Approximately 1° of latitude x 1° longitude boxes were selected along the coast to characterize
the biogeochemical properties of the source water that the Puddies eventually enclose upon formation (see
Section 2.6.1).
Over the 9-year study period in the simulated study region (Figure 1a), an average of approximately 14
Puddies were observed at each time step, resulting in a total of ~15,340 Puddy profiles (defined as the
vertical profile associated with the $\delta h_{max}$) identified (see Section 2.5). If the same eddy remained within the
same 1°x1° grid area, it was counted multiple times (using a 3-day time step) until the center of the eddy
moved to an adjacent 1°x1° grid. The area with the highest density of identified Puddies profiles was
concentrated in the coastal region (2,548), within the first ~100 km from shore, with the maximum
abundance noted between 29º – 35ºS (Figure 1b). To assess the impact of Puddies along the coastal strip, we
evaluated the mean distribution of several variables (salinity, $O_2$, $NO_3^-$, $NO_2^-$, $NH_4^+$, and $N_2O$) in areas of
approximately 1°x1° along the coast from 20°S to 38°S and between the isopycnal surfaces $S_{upper}$ and $S_{lower}$
that define the OMZ core in the model (see details in Section 2.6.1). Then, we calculated the mean profiles of
these variables in the center of the Puddies observed in each discrete coastal area (1°x1°) and estimated
anomalies of the profiles relative to the general mean profile of the corresponding box (Figures 3, see also
Figure S4 in the supplementary material).
Meridional changes in water properties along the coastal strip impact the initial properties of the Puddies. In
the northern sector (between 20°S and 30°S), the waters are warmer, more saline, and have lower $O_2$ (Figures
3a, 3d), with higher concentrations of $NO_2^-$ and $NH_4^+$ (Figures 3j and 3m). In contrast, towards the south
(south of 30°S), these characteristics generally show the opposite tendency, consistent with the water
properties observed within Puddy cores. However, $NO_3^-$ and $N_2O$ exhibit maximum levels in the central
region (near 30°S) (Figures 3g and 3p), where eddies with the highest $N_2O$ concentrations are also observed.
Oxygen levels were higher at the upper and lower limits of the OMZ (i.e., near $\sigma_\theta = 26.3$ kg m$^{-3}$ and $\sigma_\theta = 26.7$
kg m$^{-3}$) and remained relatively low in the OMZ core ($\sigma_\theta \sim 26.5$ kg m$^{-3}$). Both $NH_4^+$ and $NO_2^-$ anomalies
generated by the Puddies showed maximum values in the upper limit of the eddy cores (near $\sigma_\theta = 26.3$ kg m$^-$
$^3$; Figures 3k, 3n) and were fairly uniform along the coastal strip, except between 22º - 24ºS and the north-
central region, which showed slightly higher anomalies (Figures 3k). South of 33°S, in the narrowest part of
the OMZ, the largest anomalies in $O_2$, $NO_3^-$ and $N_2O$ are observed (Figures 3e, 3h, 3q), accompanied by a
higher standard deviation (Figures 3f, 3i, 3r).


**3.2 Biogeochemical characteristics inside the offshore Puddies**
From the total of 15,340 Puddies profiles, statistics over the 6 subregions defined in Section 2.2 (see also
Figure 1a) are performed, which is synthesized in Tables 1 to 3. Table 1 provides statistics associated with
general characteristics of the subregion, Table 2 presents statistics on ESSW properties, while Table 3
focuses on Puddies' characteristics.

**Table 1. General characteristics of the study subregions. The S$_{upper}$ layer refers to $\sigma_\theta$ = 26.0 kg m$^{-3}$, the S$_{core}$ layer refers to $\sigma_\theta$ = 26.6 kg m$^{-3}$ and the S$_{lower}$ layer corresponds to $\sigma_\theta$ = 26.9 kg m$^{-3}$. Their average depth was indicated with (±) representing the errors estimated based on a bootstrap method (see Methods). Mean thickness was defined as the distance between the S$_{upper}$ and S$_{lower}$ isopycnal layers.**

| Regions | Area (km$^2$) | Depth S$_{upper}$ (m) | Depth S$_{core}$ (m) | Depth S$_{lower}$ (m) | Mean thickness (m) |
|---------|---------------|------------------------|-----------------------|------------------------|---------------------|
| NCZ | 449 550 | 108±2 | 267±3 | 466±4 | 360 |
| NTZ | 655 300 | 149±3 | 277±2 | 449±2 | 300 |
| NOZ | 1 220 800 | 188±3 | 305±3 | 456±3 | 270 |
| SOZ | 1 808 100 | 176±9 | 289±10 | 439±10 | 260 |
| STZ | 359 760 | 151±3 | 278±3 | 451±3 | 300 |
| SCZ | 113 600 | 99±3 | 256±2 | 454±4 | 355 |

### 3.2.1 Oxygen and salinity relationship in ESSW

An evaluation of the relationship between average salinity ($< S >_{9-year}$, as a conservative variable) and average oxygen ($< O_2 >_{9-year}$) and average AOU ($< AOU >_{9-year}$) was conducted for each subregion (Table 2). This analysis was performed using the 26.6 kg m$^{-3}$, where the ESSW core was observed (see Section 2.2).

Each water mass acquires characteristics through physical and biogeochemical processes, producing particular relationships between the physicochemical variables. Low O$_2$ waters are closely related to relatively salty ESSW waters. In our study region, this water mass is located between two low-salinity and relatively well-ventilated water masses (i.e., SAAW above and AIWW below). Thus, salinity and O$_2$ show a linear inverse correlation between the upper and lower oxyclines that delimit the OMZ. Nevertheless, the occurrence of biogeochemical processes can disrupt this relationship. Therefore, it is useful to quantify which regions show these nonlinear biogeochemical processes in the context of the hypothesis that a linear relationship between O$_2$ and salinity corresponds to an aging of the water mass due to lack of ventilation. A nonlinear relationship would imply the presence of other processes, such as denitrification. Linear regression was performed between absolute salinity, O$_2$, and AOU on the S$_{core}$ surface, where AOU provides a measure of the apparent O$_2$ consumption since the ESSW formation (Table 2).

**Table 2**. **Linear regression between average AOU (µM, $<AOU>_{9-year}$) and average salinity (g kg$^{-1}$,**
$<S>_{9-year}$ **), and between average oxygen (µM, $<O_2>_{9-year}$) and $<S>_{9-year}$ within each**
**subregions  in the S$_{core}$ layer ($\sigma_\theta$ = 26.6 kg m$^{-3}$).  Linear correlation coefficient (R$^2$) provides a measure**
**of how well  $<AOU>_{9-year}$ and $<O_2>_{9-year}$ are accounted for by a linear model where mean**
**absolute salinity is considered a predictor. Error in the parameters' estimate is provided.**

| Regions | $<AOU>_{9-year}= a \times <S>_{9-year}+ b$ | | | $<O_2>_{9-year}= a' \times <S>_{9-year}+ b'$ | | |
|---|---|---|---|---|---|---|
| | R$^2$ | a | b | R$^2$ | a' | b' |
| NCZ | 0.88 | 112.5±0.57 | -3664.6±21 | 0.92 | -142±0.6 | 4963.5±20.1 |
| NTZ | 0.85 | 199.2±1 | -6685.2±33.5 | 0.88 | -229±0.9 | 7998.6±33.4 |
| NOZ | 0.96 | 380.3±0.6 | -12908±20.3 | 0.97 | -412±0.6 | 14281±20.4 |
| SOZ | 0.75 | 409.3±1.6 | -13893±55.6 | 0.78 | -445.2±1.6 | 15410±55.6 |
| STZ | 0.95 | 294.5±1 | -9989.3±36.2 | 0.96 | -327.1±1 | 11396±36 |
| SCZ | 0.98 | 200.8±0.7 | -6737±25.2 | 0.99 | -232±0.7 | 8097±25.6 |



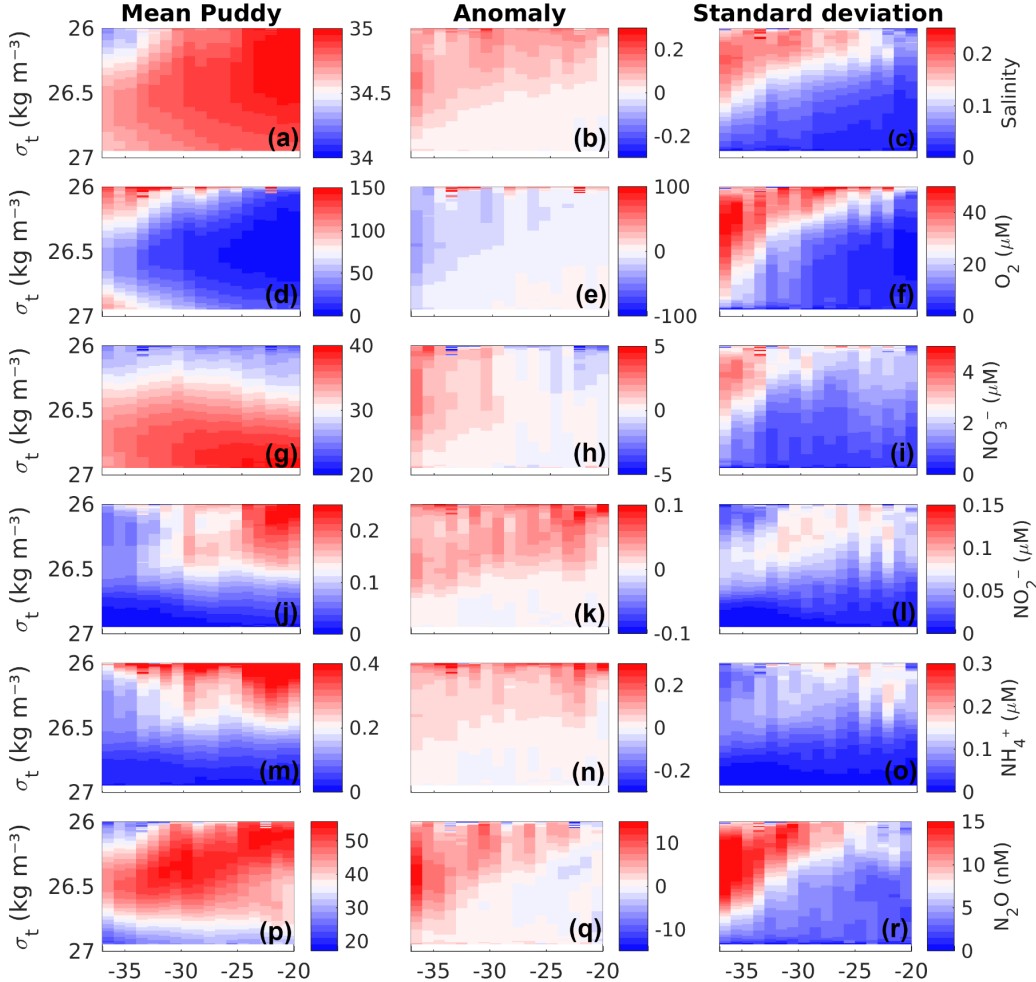

**Figure 3. Composite mean vertical profiles (left column), their anomalies relative to the long-term mean (middle column) and the standard deviation of the anomalies (right column) associated with biogeochemical features within the center of the Puddies over the first ~100 km from the slope where there are higher occurrence Puddies (strip in green in Figure 1b). From top to bottom: (a-c) absolute salinity (g kg$^{-1}$), (d-f) O$_2$, (g-i) NO$_3^-$, (j-l) NO$_2^-$, (m-o) NH$_4^+$, and (p-r) N$_2$O. Eddies' mean profiles were obtained by calculating the mean of total profiles identified during nine years in 1°x1° boxes along the coast between the isopycnal layers S$_{upper}$ and S$_{lower}$ (See Methods). The standard deviation is interpreted as the variability of the properties existing at the center. The *x*-axis represents the latitude in degrees, and the *y*-axis shows the potential density anomaly.**

Oxygen and AOU exhibited a strong linear relationship with absolute salinity (R$^2$ > 0.85) in regions where there is a greater contribution from ESSW (NCZ, NTZ, NOZ, STZ, and SCZ). The slope values ranged from 112.5 to 380 for AOU and 142 to 412 for O$_2$ fit. In contrast, the SOZ exhibited a weaker linear relationship (R$^2$ < 0.80), likely due to the mixing properties of SAAW and AAIW with ESSW (Figure 2, Table S1). SCZ (R$^2$ > 0.98) showed the strongest relationship in the coastal zone, which decreased in NCZ (R$^2$ < 0.96) where denitrification processes have been recorded and there is an evident high value of AOU and a nitrate deficiency (Figures 4a, 4b; Tables 2 and S1). In the northern region, the linear fit improves as it approaches the oceanic zone (NOZ) (R$^2$ > 0.96), indicating a strong presence of the ESSW and a reduced influence from either this biological process or mixing with another water mass.

444

### 3.2.2 Conditions of Suboxia and Hypoxia in the Oceanic Puddies

We assessed the number of Puddies exhibiting suboxia and hypoxia in each region by identifying the predominant type of low-oxygen cores in coastal, transition, and oceanic zones. The main results are presented in Table 3. The percentage of Puddies exhibiting hypoxia ($O_2 < 45$ μM) and suboxia ($O_2 < 20$ μM) was determined by classifying the range of $O_2$ concentrations observed in the center of the eddies. The presence of Puddies with these characteristics in the more remote regions was quantified (referred to as NOZ and SOZ in Table 3).

**Table 3. Percentage of total identified Puddies showing suboxic ($O_2 < 20$ μM) and hypoxic (20 μM < $O_2$ < 45 μM) cores within each region. Total profiles indicated the Puddy profiles identified over the entire study period. Mean thickness shown the distance between $S_{upper}$ ($\sigma_\theta = 26.0$ kg m$^{-3}$) and $S_{lower}$ ($\sigma_\theta = 26.9$ kg m$^{-3}$) and $\delta h$ is the average anomaly thickness generated due to the passage of Puddies in that region (i.e. composite $\delta h$ anomaly).**

| Regions | Total profiles | Mean thickness (m) | $\delta h$ (m) | Suboxic cores (%) | Hypoxic cores (%) |
|---------|---------------|--------------------|----------------|-------------------|-------------------|
| NCZ | 2,295 | 475 | 115 | 100 | 0 |
| NTZ | 2,114 | 320 | 20 | 70 | 30 |
| NOZ | 3223 | 325 | 55 | 15 | 66 |
| SOZ | 4781 | 475 | 215 | <1 | 9 |
| STZ | 1,941 | 540 | 240 | 9 | 60 |
| SCZ | 986 | 550 | 195 | 30 | 70 |



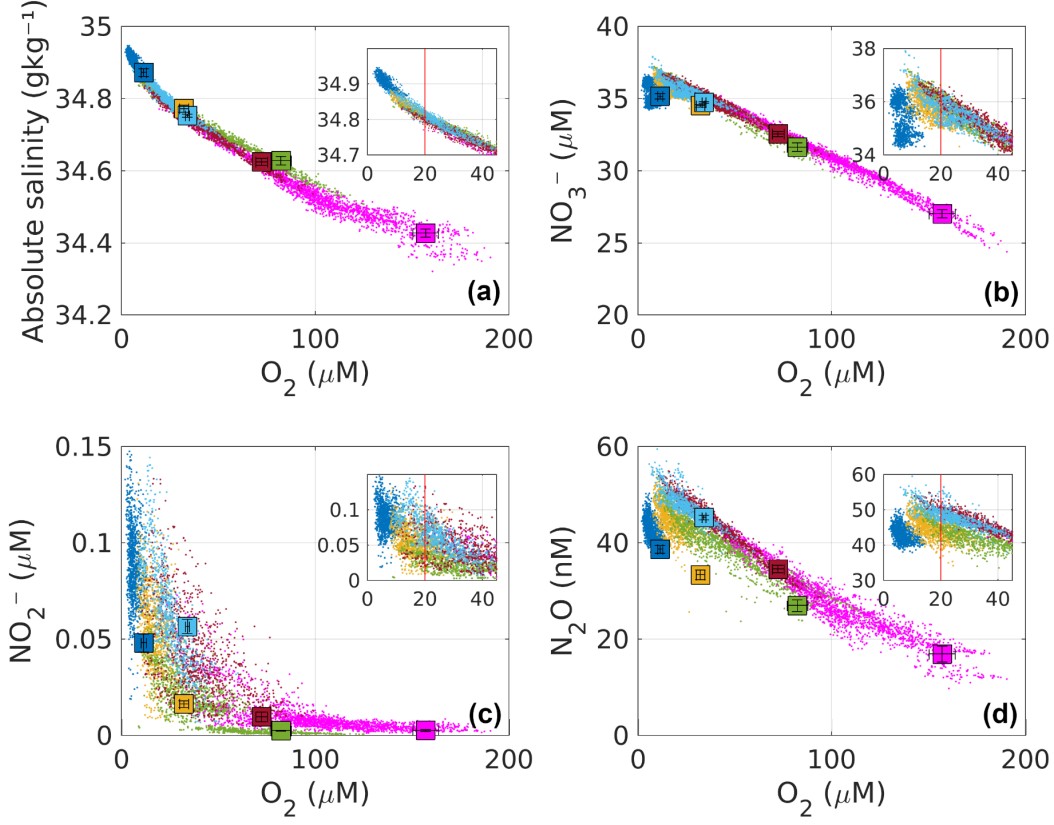


**Figure 4. Relationship between mean $O_2$ concentration and (a) absolute salinity, (b) $NO_3^-$, (c) $NO_2^-$, and (d) $N_2O$ at the $S_{core}$ layer for all identified Puddies. The dots represent the core properties of Puddies (in the $S_{core}$) of all Puddies identified over a 9-years period, at each timestep. The color indicates the subregion (see Section 2.6.1), which was compared to the mean state value (colored squares, Section 2.3). A positive oxygen gradient is observed from coastal regions (NCZ - dark blue; SCZ - light blue), followed by transition zones (NTZ - yellow; STZ - brown) to the oceanic regions (NOZ - green; SOZ - magenta). Error bars inside the squares correspond to the standard deviation of biogeochemical values within $S_{core}$ for each subregion. The smaller box shows a magnified view of the larger box for hypoxic (20 µM < $O_2$ < 45 µM) and suboxic (1 µM < $O_2$ < 20 µM) conditions to highlight . The red line indicates the threshold of $O_2$ = 20 µM.**

472

### 3.2.3 Differences between biogeochemical properties inside and outside Puddies

To understand the biogeochemical impacts of eddies in oceanic waters, the tracers' values found in the core of the Puddies were compared with the "mean conditions" (Figure S5) of each region in $S_{core}$. The presence of Puddies manifests a physical change in the mean thickness by the isopycnal layers perturbation during the passage of Puddies, which manifests itself with the average anomaly thickness ($\delta h$), different in each subregion (Table 3) and biogeochemical changes with greater contrast observed away from the coast in the south (Figure 4). Values of $< AOU' >, < \Delta NO_3' >$ and $< \Delta N_2O' >$ observed inside the Puddies were higher than $< AOU >_{9-year}, < \Delta NO_3 >_{9-year}$ and $< \Delta N_2O >_{9-year}$ observed outside (Figure 4, 5, 6; Table S1,

481  S2) confirming that eddies maintain hypoxic or suboxic cores that impact regions farther offshore, as shown
482  in the Section 3.2.2 (Table 3).

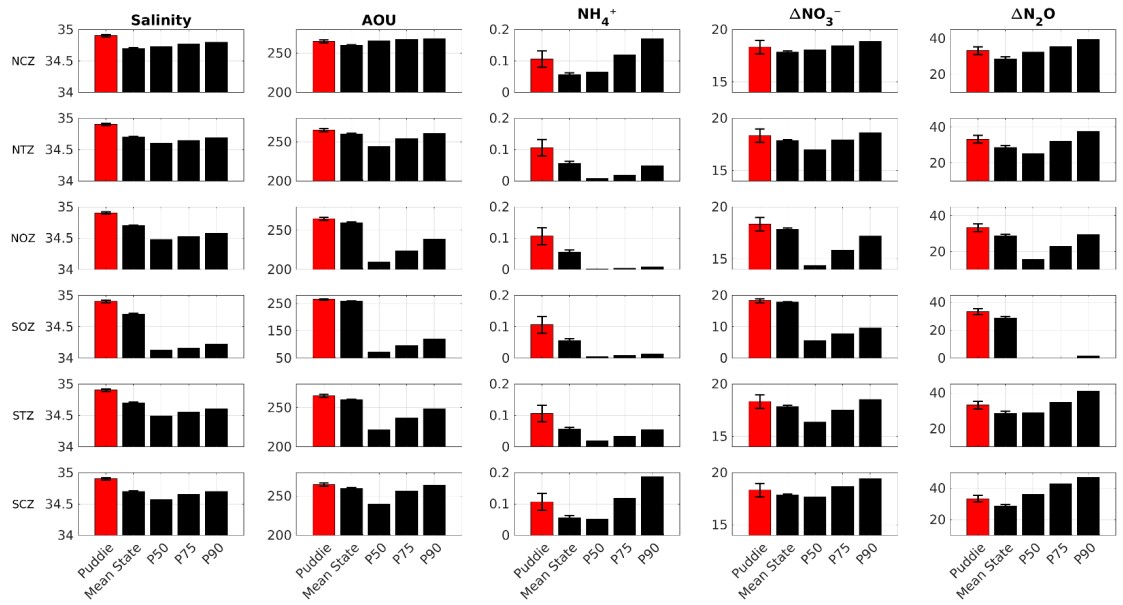

483

**Figure 5. Comparison between the average value associated with all Puddies (red color), the mean state, and the 50th, 75th, and 90th percentiles (black color, P50, P75, P90 respectively) for absolute salinity (g kg⁻¹), AOU, $NH_4^+$, $\Delta NO_3^-$ (μM), and $\Delta N_2O$ (nM) in each region (see Section 2.4). The profiles for calculating the percentiles are based on a distribution between 235 to 6800 values - according to the size of the region - obtained from random resampling using Monte Carlo's method (see Section 2.7). The averages were calculated for the $S_{core}$ layer (see Methods) . The errorbar indicates the standard deviation for Puddies and for the mean state (see also Tables S1 and S2 for complementary information).**

On the other hand, the asymptotic behavior of $NO_2^-$ was similar to that of $NH_4^+$ (not show), occurring when
$O_2 < 45$ μM (NCZ, NTZ, and SCZ) and tending towards undetectable in regions where $O_2 > 45$ μM (Figure
4c). In SCZ, Puddies with higher concentrations of $N_2O$ than in NCZ (which has suboxic conditions) and
NTZ (with hypoxic conditions) can be observed (Figure 4d, Table S1).
In general in the NCZ and SCZ, Puddies maintain the same average salinity, exhibited an increment of 0.5 -
0.7  μM $\Delta NO_3^-$ (+2.5 – 4 %), 0.01 up to 0.05 μM $NH_4^+$ similar to $NO_2^-$ (+12.5 – 83 %), and 3 – 4.5 nM $\Delta N_2O$
(+9 – 16 %) associated with an increase of  5 – 9 μM AOU (+2 – 4 %) higher than the mean state (Figures 4
and 5; more details, see Table S1). These anomalies were greater than the 50th percentile (P50) for $\Delta NO_3^-$
and $\Delta N_2O$ and close to the 75th percentile (P75) for AOU and  $NH_4^+$ with salinity exceeding P90 (Figure 5),
indicating a significant elevation of biogeochemical elements in the coastal zones due to Puddies compared
to the mean state. Offshore, the contrast with the mean state was greater, particularly in the SOZ, reaching
100% for $NH_4^+$, 43% for $\Delta NO_3^-$, 215% for $\Delta N_2O$, 45% for AOU, and 0.3  g kg⁻¹ for salinity. These values
exceed the P90 (Table S2). Similarly, in the NOZ and STZ, only $NH_4^+$ exceeded P90, while the rest remained
at P75. In NTZ, the AOU and $\Delta N_2O$ had values close to or greater than P75, while $NH_4^+$ and $\Delta NO_3^-$ exceeded
P90 (Figure 5). The perturbations to the mean state contributed by Puddies in the open sea are more
significant than near the coast, although salinity and biogeochemical tracers decrease as the core becomes
more oxygenated offshore.
The AOU/$\Delta NO_3^-$ ratio allows us to quantify the remineralization of organic matter through aerobic processes,
which is determined by the Redfield ratio ($R_{N/O}$; 16/138 = 0.11; Redfield et al., 1963). Changes in this
relationship indicate the presence of other biological processes that contribute/consume nitrogen in a system,
such as nitrogen fixation ($\Delta NO_3^-$/AOU > $R_{N/O}$) and denitrification ($\Delta NO_3^-$/AOU < $R_{N/O}$). On the other hand,
the $\Delta N_2O$/AOU ratio provides a measure of $N_2O$ accumulation, so that a high $\Delta N_2O$/AOU is associated with
the denitrification process (Sarmiento & Gruber, 2006).

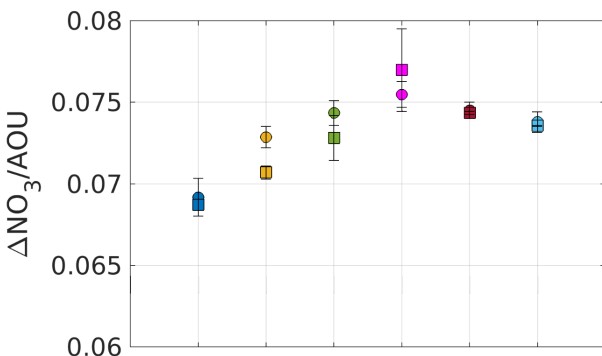

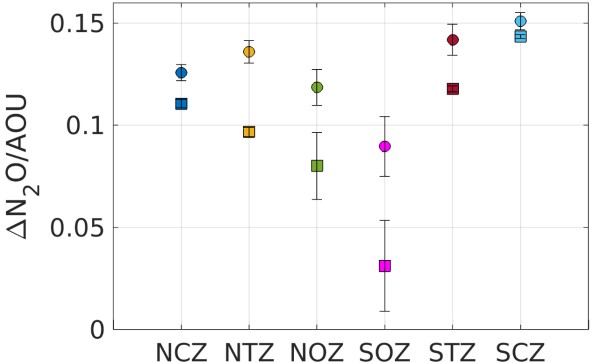


**Figure 6. Mean ratios for composite Puddies and mean conditions within each region. (Upper panel) Ratios of**

**$\Delta NO_3^-$/AOU. Note that values are smaller than the Redfield ratio ($R_{N/O}$ = 0.11). (Lower panel) Ratios of**

**$\Delta N_2O$/AOU. Values of composite Puddies are indicated with a circle while values of the mean state conditions are**

**indicated with a square. Values are for the $S_{core}$ layer (see Methods). Color code is similar to that of Figure 4. The**

**error bar indicates half the standard deviation calculated from the relationship between $\Delta NO_3^-$/AOU and**

**$\Delta N_2O$/AOU within the $S_{core}$ for each subregion .**

We observed a $\Delta NO_3^-$/AOU ratio lower than  $R_{N/O}$, inside and outside the Puddies (Figure 6, upper box),
indicating a deficit of $NO_3^-$ and high $O_2$ consumption due to the elevated remineralization in subsurface
waters. The ratio is higher in the suboxic region (NCZ) with an AOU:$\Delta NO_3^-$ = 15:1, while in the other
regions this signal of old and poorly ventilated waters also extends, albeit in different proportions (Table S1).
When $\frac{<\Delta NO_3'>}{<AOU'>} < \frac{<\Delta NO_3>_{9-year}}{<AOU>_{9-year}}$, it must be the case that $< AOU' >$ greater than $< AOU >_{9-year}$ or $<$
$\Delta NO_3' >$ is less than $< \Delta NO_3 >_{9-year}$. Since the second assumption is not met, it is true that $< AOU' >$ is
greater than $< AOU >_{9-year}$ (Table S1). However, in regions NTZ and NOZ, $\frac{<\Delta NO_3'>}{<AOU'>} > \frac{<\Delta NO_3>_{9-year}}{<AOU>_{9-year}}$,
indicating that $< \Delta NO_3' >$ greater than $< \Delta NO_3 >_{9-year}$ or $< AOU' >$ less than $< AOU >_{9-year}$. Given
that $< AOU' >$ greater than $< AOU >_{9-year}$, it must have occurred that $< \Delta NO_3' >$ is greater than $<$
$\Delta NO_3 >_{9-year}$, so the production of $NO_3^-$ was greater in the Puddies found in these regions compared to
others (Table S1). On the other hand, it holds that $\frac{<\Delta N_2O'>}{<AOU'>} > \frac{<\Delta N_2O>_{9-year}}{<AOU>_{9-year}}$. Since $< AOU' >$ greater than $<$
$AOU >_{9-year}$, it also follows that $< \Delta N_2O' >$ greater than $< \Delta N_2O >_{9-year}$, therefore the production of
$N_2O$ was proportionally larger than $O_2$ consumption in the Puddies (Figure 6, lower box; Table S1). Note that
although in SCZ and STZ the eddies exhibited the highest values of $\frac{<\Delta N_2O'>}{<AOU'>}$, $\frac{<\Delta N_2O>_{9-year}}{<AOU>_{9-year}}$ was also high
(Figure 6). Additionally, the variability of physicochemical conditions in the core increased with distance
from the coast and oxygenation.
To sum up, according to our model results, Puddies generate significant changes for the mean conditions
observed in all regions; particularly large differences were found in AOU and $\Delta N_2O$ in SOZ. In contrast, the
differences in $\Delta NO_3^-$ and AOU were not significant in the coastal regions (NCZ and SCZ).
**3.3 Oxygen Budget in the Puddies**
The processes that contribute to the modulation of the total $O_2$ content are represented in the advection-
diffusion equation (Eq. 1, Section 2.1). One part of the equation is associated with $O_2$ changes due to
physical processes (referred to as *PHYS*), and another part is related to the production/consumption of $O_2$
through biogeochemical processes (*SMS*), so we can write $\frac{\partial O_2}{\partial t} = PHYS + SMS$. PHYS comprises horizontal
advective processes (HADV = XADV + YADV), horizontal mixing (HMIX) in the *x-y* plane, vertical
advection (VADV), and vertical mixing (VMIX) in the *z*-direction so that PHYS = HADV + VADV + HMIX
+ VMIX. Note that HMIX and VMIX are mainly related to small-scale (subgrid-scale processes) mixing
(e.g., Pizarro-Koch et al., 2019). On the other hand, the SMS (source minus sink processes of $O_2$) includes
primary production, remineralization, nitrification, and zooplankton excretion (SMS = PP + Rem + Nitrif +
Exc).
We analyzed the contribution of each component of the equation to the $O_2$ balance in the center of the
Puddies, dividing the contributions above and below the $S_{core}$ (Figure 7). This was motivated by the fact that
there is a significant variance in the upper part of the eddy cores, which is also associated with a large
skewness (Figure S6). This indicates a marked vertical variability in the probability density function of the
different terms involved in the $O_2$ balance in the upper part of the eddy cores, probably induced by vertical
mixing. Positive (negative) values indicate $O_2$ increasing/production (decreasing/consumption). In general,
we observed that the contribution of PHYS was negative, mainly in the lower part of the eddy, while only
NCZ and SCZ positive values dominated at the top, although they were weak, indicating $O_2$ increasing
within the core (Figure 7c). On the other hand, $O_2$ consumption through SMS (maximum in the transition
zones NTZ and STZ) occurred in the upper part of the Puddies (Figure 7f). This proves that elevated
biological activity is maintained in the eddies far from shore, with higher intensity in younger eddies but
weakening in eddies that reach regions very distant from the coast (NOZ and SOZ). Although the SMS
fluxes are of the order $O(10^{-6})$, one order of magnitude lower than the PHYS, small changes in $O_2$ due to
SMS can also result in a sharp local spatial gradient in $O_2$ that induces strong changes in the advection of $O_2$.
This could significantly impact on the behavior of biogeochemical components that are highly sensitive to
minimal changes in $O_2$ concentration, especially under hypoxic or suboxic conditions.

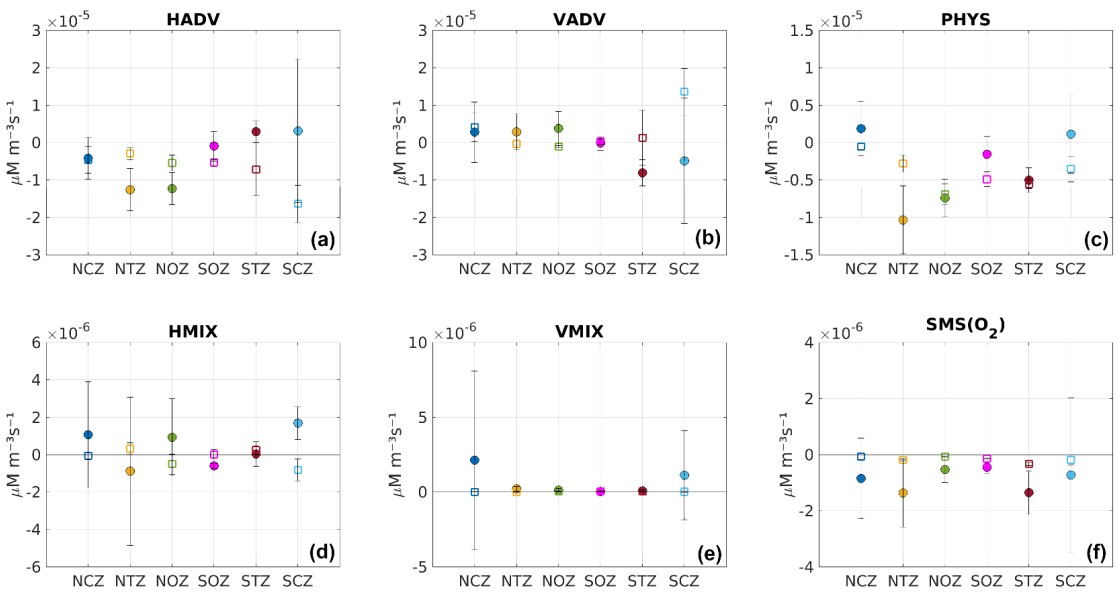


**Figure 7. Mean terms involved in the $O_2$ budget (see equation 1). The different terms were temporally averaged inside the eddies in each subregion. PHYS is the sum of physical processes, which encompasses horizontal advection (HADV = XADV + YADV), vertical advection (VADV), horizontal mixing (HMIX), and vertical mixing (VMIX). The biogeochemical processes are included as sources minus sinks (SMS) processes impacting $O_2$. In particular, SMS includes the $O_2$ fluxes by primary production, nitrification, remineralization, and excretion of the zooplankton. Positive (negative) values indicate processes contributing to increasing (depleting) $O_2$ inside the eddy. (a) HADV (b) VADV, (c) PHYS, (d) HMIX, (e) VMIX, and (f) SMS. The averaging was carried out by dividing the composite Puddy profile (see Section 2.6.1) into two parts: above (fill circles) and below (squares) the 26.5 kg/m³ isopycnal surface. This division was based on the marked vertical variability observed in the probability density function, as indicated by the variance and skewness (Figure S6). The vertical bars represent standard deviations of the $O_2$ budget terms within $S_{core}$ for each subregion.**

The advection components $(O(10^{-5}))$ can be interpreted as the ability to maintain $O_2$ at the center of the eddy
during the eddy's displacement. They dominate the $O_2$ budget compared to mixing processes $(O(10^{-6}))$
involving diffusion of $O_2$ (Figures 7a, 7b). The lateral fluxes (HADV) showed $O_2$ leakage (HADV < 0) from
the core mainly in the northern eddies but not in the southern eddies (HADV > 0), where $O_2$ influx to the
core was evident (Figure 7a). Assuming vertical oxygen advection dominates over shear-induced oxygen
transport, in the northern regions, VADV > 0 dominates in the upper part of the Puddies, indicating $O_2$ input.
In contrast, in the southern regions, this pattern is reversed. In coastal zones, vertical advective processes
become significant, with VADV > 0, highlighting the greater $O_2$ input in SCZ (Figure 7b).

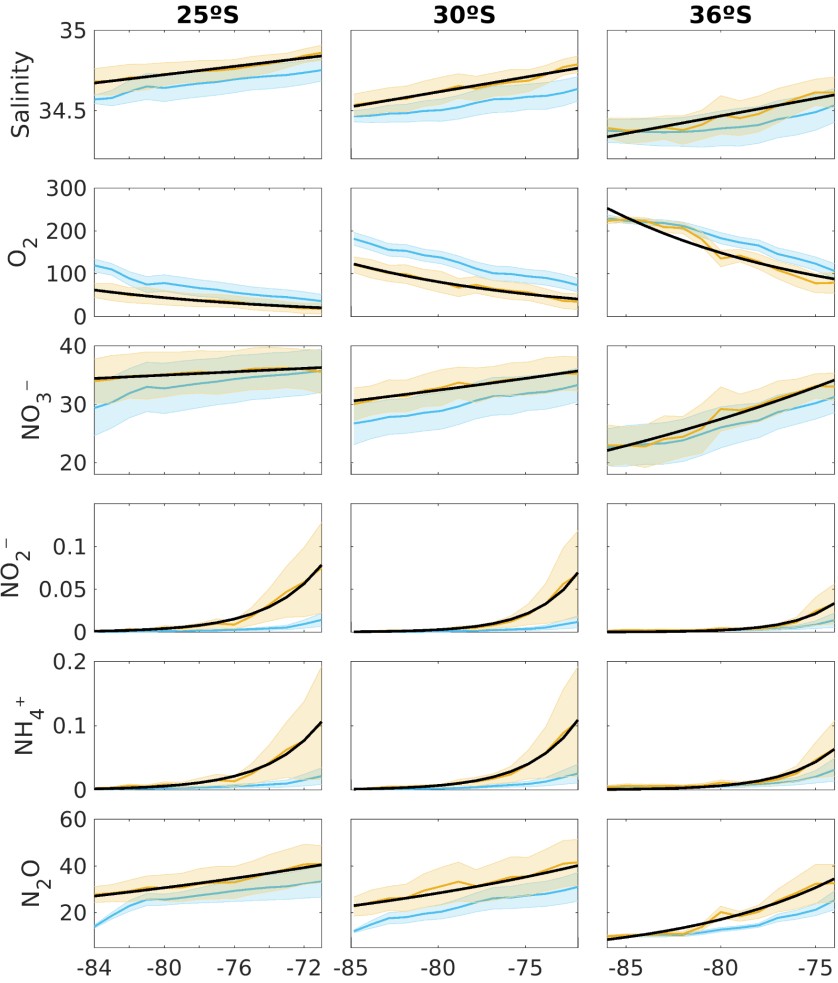


**Figure 8. (orange line) Average biogeochemical properties of the total volume of Puddies identified in the zonal**
**bands corresponding to the latitudes of 25°S (left), 30°S (middle), and 36°S (right),  and (blue line) the mean state**
**associated with that volume . In order from top to bottom: absolute salinity (g kg$^{-1}$), oxygen, nitrate, nitrite,**
**ammonium ($\mu$M) and nitrous oxide (nM). The *x*-axis shows the longitude in degrees west.  At 25°S, 30°S and 36°S**
**862, 810 and 658 Puddies profiles were considered, respectively.  The shading indicates the standard deviation**
**amongst the Puddies. The black line stands for the result of  an exponential model of the form y = Ae$^{-kx}$ that**
**provides  the decay rate of mean biogeochemical properties as the distance from the coastal zone (see Table 4 for**
**values of *k* for both mean state and Puddies).**

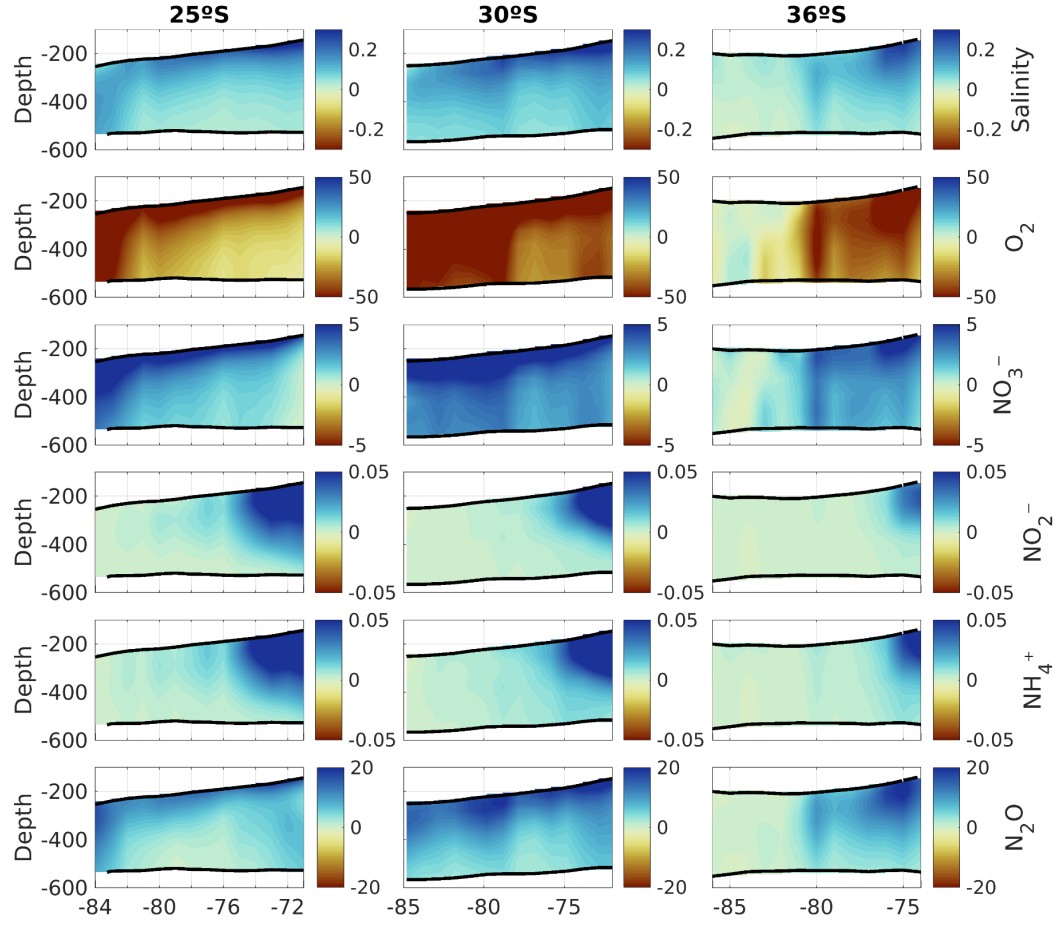


**Figure 9. Puddies composite anomalies for tracers along three zonal bands at different latitudes (25ºS - left, 30ºS -**
**middle and 36ºS - right). In order from top to bottom: absolute salinity (g kg$^{-1}$), oxygen, nitrate, nitrite,**
**ammonium (μM) and nitrous oxide (nM). Anomalies are only shown between the isopycnals layers S$_{upper}$ and**
**S$_{lower}$ (black contours). The *x*-axis shows the longitude in degrees. The number of Puddies used for the composites**
**is identical to that used for Figure 8.**

Vertical mixing fluxes were positive in newly formed eddies (mainly in coastal regions), while lateral
diffusion fluxes showed high variability in magnitude and direction (Figures 7d, 7e).

**3.4 Subsurface water band changes by Puddies**
In this section, a robust approach was used to demonstrate the effect of Puddies on the mean state through the
modification of biogeochemical properties, vertical structure, and the contribution of various processes to
changes in the subsurface water band delimited by the density surfaces S$_{upper}$ and S$_{lower}$ across different
latitudes (25°S, 30°S, and 36°S) during their interaction. All Puddies identified in the three latitudinal bands
over the 9-year period were used to build a composite by extracting properties in x-y-z dimensions to obtain
the average zonal variables associated with Puddies (details in Section 2.6.2).

We compared the characteristics acquired by Puddies in the subsurface band corresponding to 25°S, 30°S, and 36°S with the mean state of the same volume (Figure 8). First, we observed a decrease in salinity farther from the coast, which indicates the intrusion of external waters distinct from the ESSW. At 25°S, Puddies farther from the coast retain saltier water with higher nitrate concentrations (16% higher than mean state) and lower oxygen levels (49% less than mean state) than those at 30°S (12% increase in $NO_3^-$ and 33% decrease in $O_2$) and 36°S (which becomes completely mixed). Decay rates in Puddies at 30°S are higher than the mean state, except for $NO_3^-$ and $N_2O$, while at 36°S, only $N_2O$ exhibits a lower decay rate in Puddies (Table 4). At 25°S, the opposite is observed, with lower decay rates in Puddies except for $NH_4^+$ and $NO_3^-$. However, the biogeochemical concentrations in both Puddies and the mean state gradually decrease southward. Puddies' contributions are generally higher at 30°S, as shown by anomalies highlighting low $O_2$ and high $NO_3^-$ and $N_2O$ concentrations (Figure 9). At 25°S, Puddies input more $NH_4^+$ and $NO_2^-$ (~400% higher than the mean state) in the subsurface band near the coastal zone, while in the oceanic zone, $NO_3^-$ and $N_2O$ (~200% more than the mean state) are the primary contributions, alongside waters with low $O_2$. The maximum contribution of $NO_2^-$ occurred at 30°S, with a 460% increase compared to the mean state near the coastal zone. At 36°S, Puddies show increased contributions in the transition zones, unlike at other latitudes, with salinity, $NO_2^-$, and $NO_3^-$ maintaining high concentrations up to 76°W. $N_2O$ also exhibits higher contributions from Puddies at 36°S, particularly in the near-coastal (29% higher than the mean state) and transitional zones. $NH_4^+$ and $NO_2^-$ concentrations are highest near the formation zones at all three latitudes and decrease sharply in the transitional zones or at the beginning of the oceanic zone. In the northern regions, where the OMZ is broader, and there is less differentiation between water masses, in contrast, farther south, the SAAW erodes the OMZ, making it narrower and causing more abrupt biogeochemical changes between the coastal and transition zones.

Additionally, the dynamics associated with each latitude could drive different changes. The net oxygen budget, referred to as O-RATE in Figure 10, shows that in the upper parts of the eddies, there is always a positive oxygen contribution into the Puddies, with ventilation increasing in the southern eddies within the transitional zone, as seen at 36°S. Oxygen primarily enters through physical processes, with horizontal and vertical advection (HADV and VADV) being dominant (Figure 11). Oxygen depletion in Puddies is also influenced by advective processes, which allow the inflow of less oxygenated waters, ventilating the cores primarily in the middle and lower parts of the eddies (30°S band) or distributed between the $S_{upper}$ and $S_{lower}$ layers in the near-coastal zone (36°S band). Oceanic regions (up to ~800 km offshore) in the 30°S and 36°S bands show intensified SMS activity (e.g., ammonification, zooplankton excretion, and nitrification; Figures 10 and 11). To estimate the time Puddies take to cover a certain distance (i.e., 100 km (~1°)), we could consider a propagation speed of ~2 km/day similar to surface eddies (Chaigneau et al., 2009; Hormazabal et al., 2013). Accordingly, it would take them approximately 50 days to travel that distance. Then, the observed SMS < 0 in the oceanic zone at 30°S remained in the Puddies for about a year.

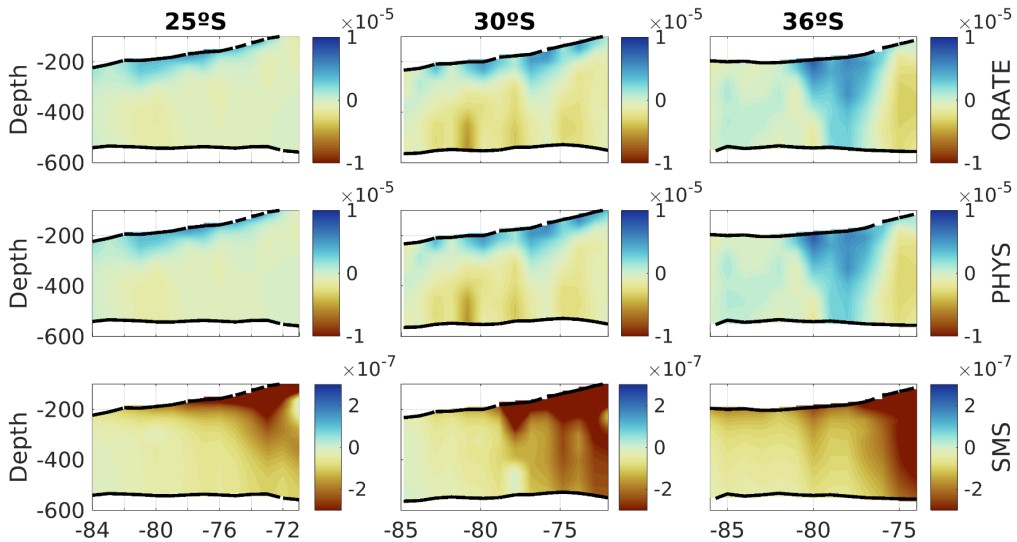


**Figure 10. Same as figure 9 but for the main terms of the O₂ budget: (top panels) the rate of change of oxygen,**
**(middle panels) the sum of all physical processes, and (bottom panels) sources minus sinks (SMS) of**
**biogeochemical processes.**

**Table 4. Estimates of the decay rate of tracers along 25°S, 30°S and 36°S for mean state and anomalous conditions**
**within Puddies. The decay rate derived from an exponential fit of the form $y = Ae^{-kx}$, where $x$ represents the**
**distance from the coast in km and $y$ is the concentration of the biogeochemical tracer. $k$ is the decay rate and $A$ the**
**tracer concentration in the coastal strip (1°x1°). A zonal extent of 1,445 km was used at 25°S and 30°S, while at**
**36°S we used 1,334 km (i.e. same range of longitude as in Figure 8). Values of k are statistically significant at the**
**95% confidence level.**

**Decay rate ($k$) using exponential fit $y = Ae^{-kx}$**

| Variable | Puddies (25ºS) | Mean state (25ºS) | Puddies (30ºS) | Mean state (30ºS) | Puddies (36ºS) | Mean state (36ºS) |
|---|---|---|---|---|---|---|
| **Salinity** | $3.35 \times 10^{-6}$ | $3.38 \times 10^{-6}$ | $4.8 \times 10^{-6}$ | $3.5 \times 10^{-6}$ | $5.7 \times 10^{-6}$ | $3.3 \times 10^{-6}$ |
| **NO₃⁻** | $4 \times 10^{-5}$ | $1 \times 10^{-4}$ | $1 \times 10^{-4}$ | $1.6 \times 10^{-4}$ | $3.2 \times 10^{-4}$ | $2.6 \times 10^{-4}$ |
| **NO₂⁻** | $3 \times 10^{-3}$ | $3 \times 10^{-3}$ | $4 \times 10^{-3}$ | $3 \times 10^{-3}$ | $4 \times 0^{-3}$ | $3 \times 10^{-3}$ |
| **NH₄⁺** | $3 \times 10^{-3}$ | $2 \times 10^{-3}$ | $3.1 \times 10^{-3}$ | $2.8 \times 10^{-3}$ | $3 \times 10^{-3}$ | $2 \times 10^{-3}$ |
| **O₂** | $-7.6 \times 10^{-4}$ | $-8 \times 10^{-4}$ | $-7.7 \times 10^{-4}$ | $-6 \times 10^{-4}$ | $-7.9 \times 10^{-4}$ | $-5 \times 10^{-4}$ |
| **N₂O** | $3 \times 10^{-4}$ | $4 \times 10^{-4}$ | $4 \times 10^{-4}$ | $5 \times 10^{-4}$ | $1 \times 10^{-3}$ | $8 \times 10^{-4}$ |


At 30°S and 36°S, SMS < 0 extends into the lower parts of the eddies near the coastal zone more intensely
(~100 - 150 days), potentially prolonging the lifespan of $NO_2^-$ and $NH_4^+$. Furthermore, lower oxygen
concentrations intrude through HADV and VADV, particularly at 36°S.

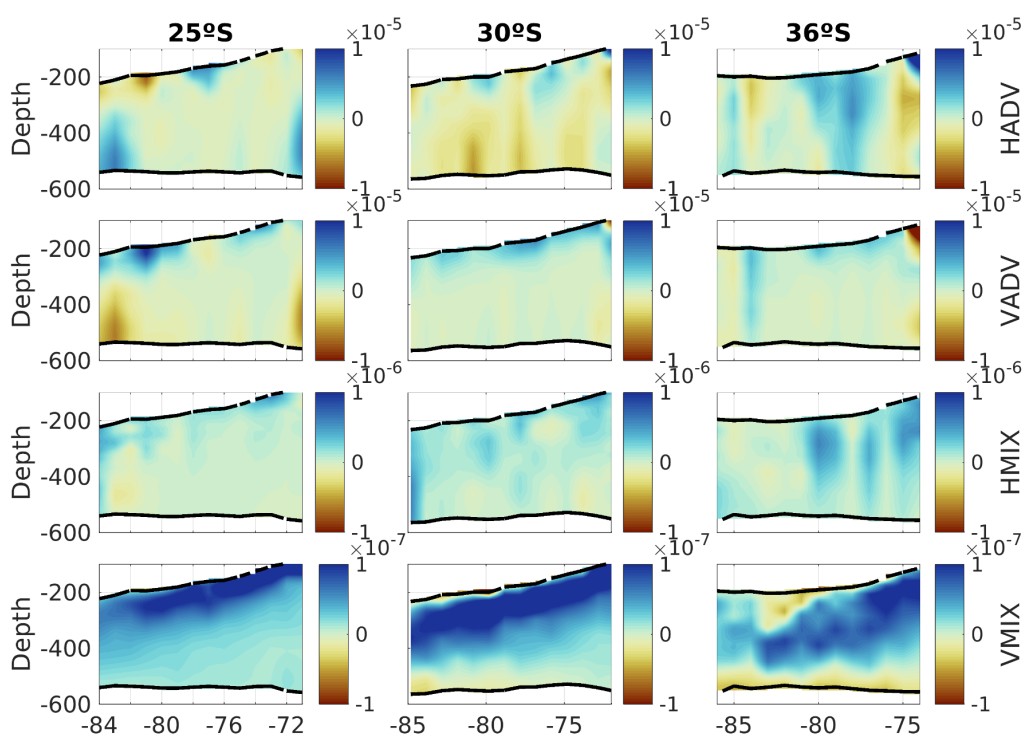

**Figure 11. Same as figure 10 but for the various contributions of PHYS (see Eq. 1).  from top to bottom:**
**Horizontal advection (HADV), vertical advection (VADV), horizontal mixing (HMIX), vertical mixing (VMIX).**
**4 Discussion**
Several studies have reported *in situ* subsurface eddies with similar biogeochemical characteristics to those
observed in the present study (Stramma et al., 2013; Stramma et al., 2014; Cornejo D'Ottone et al., 2016;
Arévalo-Martínez et al., 2016; Grundle et al., 2017; Hormazabal et al., 2013; Kartensen et al., 2017).
However, studies focused on a detailed recycling of bioelements within these eddies are scarce. While recent
studies have used high-resolution coupled models to describe and quantify more complex processes involved
in the $O_2$, nutrient, or organic matter balance within eddies (José et al., 2017; Frenger et al., 2018; Loveccio
et al., 2022), the agents generating natural variability in the lifespan of these bioelements associated with the
nitrogen cycle during their trajectory from the OMZ to oceanic waters, have not been analyzed in detail.
In this work, a robust statistical analysis was conducted considering all Puddies identified over a 9-year
period using two approaches. In the first approach, we characterized the biogeochemical signal of the source
water inside the Puddies. Subsequently, we analyzed the eddy cores, focusing on the average Puddies'
profiles (using $\delta h_{max}$; see Section 2.6.1) and presented the values in the isopycnal layer of 26.6 kg m⁻³ to
assess how and to what extent the biogeochemical properties of the Puddies' cores differ across various
regions of the SEP from their formation zone (results in Sections 3.1, 3.2, and 3.3).
In the second approach, we considered the entire volume of the Puddies to characterize their average
conditions and quantify how these properties modify the intrathermocline band where the Puddies travel—
from the coastal zone, through the transition zone, and into the oceanic zone (see Section 3.4). Additionally,
we evaluated the balance of processes involved in the interaction of the Puddies along their trajectory (details
in Sections 3.3 and 3.4). This analysis allowed us to identify differences in the impact of Puddies at different
latitudes and gain a deeper understanding of the processes involved.

### 4.1 Biogeochemical anomalies in Puddies from their formation

Subsurface anticyclonic eddies appear to be formed by separating the Peru-Chile Undercurrent from the
slope (e.g., Molemaker et al., 2015; Thomsen et al., 2016; Contreras et al., 2019). Here, we observed a higher
recurrence of Puddies in specific sectors, namely, between 29º-35ºS (Figure 1b), related to the widening of
the continental shelf. This change in topography may lead to greater separation of the PCUC from the slope,
creating favorable conditions for the generation of Puddies, as observed by Chaigneau et al. (2009). Puddies
appear to originate from instabilities forced in the bottom boundary layer at the upper continental slope
where the core of the PCUC interacts with the sea bottom (Contreras et al., 2019), and where the core of the
OMZ is observed. This water, trapped within the Puddies, is characterized by positive $NH_4^+$, $NO_3^-$, $NO_2^-$ and
$N_2O$ anomalies and very low $O_2$ that vary with latitude, as described in Section 3.1. However, unlike $O_2$ and
salinity, which exhibit a more uniform gradient along the coast, several other biogeochemical components
have more irregular spatial distributions depending on whether conditions are hypoxic or suboxic.
In the northern part of the coastal zone, limited ventilation creates an environment with suboxic conditions
and denitrification, resulting in $NO_3^-$ and $N_2O$ deficits, and $NO_2^-$ and $NH_4^+$ enrichment. In contrast, the
central coastal waters exhibit increased $NO_3^-$, with the southern part showing higher $N_2O$ concentrations.
Additionally, as the formation of Puddies is associated with cross-shore velocities – exchanging nutrients
between the continental shelf and open sea (Thompsen et al., 2015) – the mixture of ESSW and SAAW
waters, also affected by the southward reduction in the contribution of ESSW (Figure 2; Silva et al., 2009),
enhances the biogeochemical variability of the Puddies generated along the Chilean coast (Figure 3).
In summary, Puddies formed along the Chilean coast capture distinct biogeochemical "signatures" depending
on where they form. In the north, minimal ventilation fosters suboxic conditions and denitrification –leading
to deficits of $NO_3^-$ and $N_2O$ but high $NO_2^-$ and $NH_4^+$– whereas central and southern subregions show
increased $NO_3^-$ and higher $N_2O$. Moreover, cross-shore exchange between ESSW and SAAW further
amplifies this variability, giving rise to eddies with diverse nutrient and oxygen properties as they move
offshore.

**4.2 Persistence of Biogeochemical Anomalies Away from the Coast**

In each subregion (Figure 1a), the Puddies exhibited distinct biogeochemical configurations, as summarized in Figures 4, 5, and 6 (results in Section 3.2). The dominance of SAAW in the southern regions explains the largest number of hypoxic Puddies containing higher $O_2$ concentrations and lower levels of $NO_3^-$, $NH_4^+$, $NO_2^-$, and $N_2O$ compared to Puddies farther north, where suboxic cores are more prevalent (Silva et al., 2009). Consequently, Puddies that capture a larger fraction of ESSW during their formation exhibit biogeochemical contributions closer to the P75 (Figure 5, Table S2) and show a higher $O_2$ deficit than the mean state. This suggests that the southern regions are experiencing greater deoxygenation and nutrient enrichment due to the influence of Puddies compared to the northern regions, where the OMZ is much broader.

When estimating the percentage of core distributions exhibiting hypoxia and suboxia, the results showed that hypoxic cores predominantly occur in oceanic waters (Table 2). In general, as we move away from the Chilean coast, biogeochemical element concentrations decrease significantly, though the magnitude of the decrease varies by element (Figure 4). Although Puddies share common characteristics within each region, some variability is observed. This reflects the variety of drivers - such as kinetic and potential energy, vorticity and other dynamics - as well as the differences in the properties of the source water it encloses, its trajectory (potentially interacting with different external water masses), and the physical and biogeochemical processes occurring within it during its lifespan. These factors can enhance or diminish the Puddy's ability to transport properties far from its formation zone.

However, the results indicate an effective transport of these properties to oceanic regions, where values exceed the P90. In contrast, for the transition zones (NTZ and STZ), only AOU, salinity, and $NH_4^+$ exceed the P90, while $\Delta N_2O$ and $\Delta NO_3^-$ are closer to the P75. This regional difference clearly demonstrates that the perturbations to the mean state caused by Puddies are more significant in the open ocean than near the coast (Figures 4 and 5; Table S1).

**4.3 Changes in the Biogeochemistry of Subsurface Water Masses by the Puddies**

Our results showed changes from the coastal to oceanic zones across three latitudes, where a decrease in dissolved inorganic nitrogen compounds, along with a salinity decay rate, confirmed that Puddies age and mix with external waters as they move into the oceanic zone (Figure 8; Table 4). Similar patterns were observed by José et al. (2017) in the upwelling system off Peru, Frenger et al. (2018) across the main EBUS, and Loveccio et al. (2022) in the Northern Canary upwelling system. This offshore change suggests that Puddies have relatively permeable boundaries, but the level of coherence and isolation within the Puddy core can significantly influence the lifespan of compounds enclosed in it. The macronutrient behavior observed was consistent with Frenger et al. (2018), though in our case, $N_2O$ —despite its minimal concentrations— exhibited a pattern similar to other macronutrients, such as $NO_3^-$ (Figure 8). Decay rates for $NO_2^-$ and $NH_4^+$ were faster than those for $NO_3^-$ and $N_2O$ (Table 4).

Nitrate showed higher concentrations compared to other nitrogen compounds. Additionally, as long as $NH_4^+$
and $NO_2^-$ are present (Figures 4, 8, and 9), nitrification can replenish the $NO_3^-$ pool in subsurface waters
where photosynthesis does not consume it. In addition, $N_2O$ production depends on $[NH_4^+]$, $[NO_2^-]$, and $[O_2]$
levels and the nitrification and denitrification processes. According to the model parameterization (for more
details, see Gutknecht et al., 2013a), the maximum $N_2O$ production occurs when $O_2 = 1$ μM, with $SMS(N_2O)$
$= \alpha(\text{Nitrif}) + \beta(\text{Nitrif})$ ($\alpha, \beta$ constant), whereas when $O_2$ is high or $O_2 < 1$ μM, the $N_2O$ production decreased
because $SMS(N_2O) \rightarrow \alpha(\text{Nitrif})$. On one hand, $N_2O$ production diminishes as $NO_2^-$ and $NH_4^+$ availability
declines, as observed in the open sea; on the other hand, in coastal zones, denitrification rapidly depletes
these compounds until $N_2O$ production ceases at oxygen concentrations below 1 μM. Since the model does
not account for $N_2O$ consumption by biological processes (via denitrification or fixation) but only through
atmospheric exchange, the Puddies act as a source of $N_2O$ as they move offshore and the observed $N_2O$
decrease within Puddies can only result from physical processes facilitating exchange with external waters.
Notably, the 30°S zonal band exhibited the largest anomalies in biogeochemical characteristics (Figure 9).
Along this band, the high occurrence of Puddies, including those formed at or south of this latitude, is
significant due to the characteristic northwest trajectory of eddies. This could explain the high nutrient
concentrations there (Figure 3). Newly formed Puddies must be supported by dissolved organic nitrogen
from the source water with higher $O_2$ consumption (*SMS*) during the first 100-150 days, as Loveccio et al.
(2022) proposed. Mixing with external waters (Figure 8) is driven by physical processes, particularly
advection (HADV and VADV, Figures 10 and 11). Compared to the other two latitudes, higher $O_2$
consumption (SMS < 0) at 30°S suggests higher local nutrient production through remineralization (i.e.,
more organic material is being broken down), potentially extending the lifespan of $NO_2^-$ and $NH_4^+$, necessary
for the $\Delta N_2O$ and $\Delta NO_3^-$, activity that remained in the Puddies for about a year around 800 km offshore.
Biological activity sustains low-oxygen cores for longer periods, preserving original conditions even with $O_2$
< 20 μM in the cores, where denitrification continues while the edges ventilate. Additionally, horizontal
advective processes further reduce oxygen levels in the lower part of the eddy by allowing the intrusion of
low-oxygen waters, contrasting with observations at 25°S and 36°S.
This combination of conditions suggests that regions of significant interaction with Puddies can transform the
biogeochemistry of the subsurface layer in meaningful ways. Both approaches used to characterize Puddies
provide valuable insights into how and to what extent these Puddies evolve, driving changes in the
intrathermocline from their formation to their dispersion across the SEP.
**4.4 Advantages and Disadvantages of the Model for Low Oxygen Conditions**
We now discuss the limitations of the model formulation in accounting for the transition between the two
oxygen regimes (hypoxic versus suboxic) and its implication for Puddies life cycle. AOU:$NO_3^-$ ratios of
250:30 (up to 20:1) have been documented in the Atlantic from *in situ* monitoring of $NO_3^-$ within mode water
eddies (Kartensen et al., 2017) and in eddy cores with $NO_3^- < 25$ μM at $O_2 < 5$ μM in the SEP (Stramma et
al., 2013). The eddies modeled here showed AOU values similar to those found by Karstensen et al. (2017)
in the eastern tropical North Atlantic, although $NO_3^-$ was overestimated by 3 - 5 μM leading to an
AOU:$\Delta NO_3^-$ ratio of 5:1 (Figures 4b and 6 - upper box; Stramma et al., 2013). Suboxic Puddies build up
$NO_2^-$, although the modeled maximum concentration appears underestimated compared to previously
reported *in situ* data (Stramma et al., 2013; Cornejo et al., 2012; Cornejo-D'Ottone et al., 2016).
Consequently, the relationship between $NO_2^-$ and $\Delta N_2O$ in the northern zone also showed underestimation.
At the modeled $N_2O$ maximum (>30 nM, Figure 6), field data shows $NO_2^-$ values >1 μM (Cornejo et al.,
2012), while the modeled $NO_2^-$ maximum reaches 0.15 μM (Figures 4 and 5). Comparing our results with
eddies monitored in the SEP, the $N_2O$ concentrations observed in open ocean eddies agree with those
measured by Cornejo-D'Otonne et al. (2016; Figure 4d, S5) in an eddy originated near the coast off
Concepcion (~37°S, south zone), although there are differences of up to 20 nM with the eddy reported by
Arévalo-Martínez et al. (2016; Figure 4d, S5) off northern Chile. These results suggest a better representation
of biogeochemical processes by the model in a hypoxic than suboxic environment.
Biogeochemical components are challenging to model due to the numerous physical, biogeochemical, and
biological processes involved. Specifically, under hypoxic or suboxic conditions, the nitrogen cycle is more
complex due to additional processes that occur within a narrow $O_2$ range, increasing the system's sensitivity.
Processes removing nitrogen from the system, such as denitrification, also generate intermediate products
such as $N_2O$ and $NO_2^-$ (e.g., denitrification, $NO_3^-$ reduction). Denitrification is a complex process to
parameterize, involving a range of steps for each nitrogen component and various rates of decomposition of
particulate (large and small) and dissolved material. However, despite the attempt to consider these processes
realistically, the model remains an approximation but reasonably represents the involved processes, primarily
within the intrathermocline band (see Appendix). In the suboxic zone, while the model underestimated $O_2$,
$NO_3^-$ remained overestimated, but in the subsurface band associated with the OMZ core, the model
represented the lowest biases in $O_2$ and nitrogen evaluated with climatological observations (see Appendix).
This allowed us to present a robust statistical analysis of biogeochemical properties inside and outside the
Puddies. Despite the difficulty in simulating realistically sharp mean gradients in water mass properties
(Pizarro-Koth et al., 2019), the model did highlight typical features, such as $NO_3^-$ consumption and
production of $NO_2^-$ and $N_2O$, processes that persist within Puddies far from their origin within the OMZ.

**5 Conclusions**
Using a high-resolution coupled simulation of the SEP, we characterized the biogeochemical changes within
Puddies, their differences with the external properties, and the processes involved in the $O_2$ balance during
their transport to oceanic waters. The model resolved eddy dynamics and biogeochemical processes related to
the nitrogen cycle, including characteristic EBUS processes such as denitrification. This methodology
allowed us to statistically approximate the biogeochemical changes occurring within these Puddies, quantify
the modifications to water bodies along their paths, and identify the dominant mechanisms regulating
compound concentrations from the formation zone to hundreds of kilometers offshore.
During formation, Puddies capture a biogeochemical signal that varies depending on their origin, which is
associated with the core of the Peru-Chile Undercurrent. Peripheral permeability enables exchange with
external waters, modulating the original signature. However, the core signal retains characteristic negative $O_2$
anomalies and positive anomalies for other biogeochemical tracers. These disturbances may cause average
properties to exceed the 90th percentile (P90) in the open ocean, in contrast to the formation zone, where
values remain above the 50th percentile (P50). While a high percentage of Puddies near the coast exhibit
suboxic cores (all in the north and 70% in central-south Chile), the proportion decreases with distance from
the coast, where hypoxic cores become predominant (60% in the STZ and NOZ). This suggests core
ventilation during their trajectory, as indicated by an estimate of the salinity decay rate as a function of
latitude.
The dominant mechanism for $O_2$ exchange in the eddy core is lateral and vertical advection, with vertical
mixing contributing two orders of magnitude less. $O_2$ consumption through biological activity (SMS) was
maintained inside the Puddies (around 6 to 12 months) up to the oceanic zones (e.g., at 30°S  the $O_2$
consumption was maintained up to  ~800 km from the coast), enabling the persistence of low $O_2$ conditions
in the core despite peripheral ventilation. This persistence supports processes such as denitrification, which
occur under hypoxic or suboxic conditions within the OMZ, and can extend into areas far beyond the OMZ,
sustained in core niches with $O_2 < 20$ μM. The SCZ and STZ experience significant deoxygenation and
nutrient enrichment due to Puddies formed with larger anomalies. At 30°S, a combination of factors
significantly impacts the intrathermocline biogeochemistry due to Puddies transiting this latitude. The
maximum contribution of $NO_2^-$ occurred at 30ºS, with a 460% increase compared to the mean state, near the
coastal zone. Clearly, the formation of Puddies is a critical process in extending the OMZ boundaries at its
southern edge. Additionally, the interplay of physical conditions can generate regions with substantial
subsurface water mass changes due to interactions with Puddies.
Biogeochemical tracers exhibit varying lifespans depending on their original concentrations, as well as their
demand and production rates. Ammonium ($NH_4^+$) and nitrite ($NO_2^-$) tend to decline first, whose
concentration was maximum in the coastal zone (300-400% higher than mean state), with high decay rates
with distance from the coast, while other components decrease more slowly and persist longer. Despite being
produced in low concentrations, $N_2O$ is retained in Puddies farther from the coast, serving as a proxy for
denitrification in the source water and contributing to the subsurface reservoir of this greenhouse gas in the
open ocean (up to ~100% higher than the mean state). Characterizing Puddies provides valuable insights into
how they evolve and drive intrathermocline changes from their formation to dispersion across the SEP.
Beyond offering insights into the biogeochemical dynamics of Puddies, this study's findings can also help
guide *in situ* monitoring, particularly in remote areas where unusual biogeochemical signals may indicate the
presence of a Puddy.
Higher-resolution models enable better characterization of mixing processes involved in eddy core
ventilation and the exchange of properties with external waters through submesoscale dynamics (Brannigan
et al. 2017). Coupling with more complex biogeochemical models will help quantify the effects of additional
SMS processes within Puddies not included in this study. This is planned for future research. Validation and
supplementary statistics through further observations in the study area are also recommended to deepen our
understanding of coupled processes in surface and subsurface eddies.

**Appendix: model assessment**

**A.1 Data**

Observed climatological $O_2$ and $NO_3^-$ fields were taken from the CSIRO Atlas of Regional Seas (CARS2009; www.marine.csiro.au/~dunn/cars2009/) for the biogeochemical model assessment, which has a spatial resolution of 0.5 degrees and a monthly mean resolution. The model validation was performed over the first 800 m. The vertical resolution of the data set is 10 meters to the first 300 m, 25 meters to 500 m, and 50 meters down to 800m.

**A.2 Surface Nitrate**

The model simulates low $NO_3^-$ (< 2 μM) in general, except in the areas near the coast and in the southwest where concentrations reached 8 μM (Fig. A1a). The observed concentrations were higher, particularly in the south (Fig. A1b). Overall, the model underestimated $NO_3^-$ concentrations in most of the domain, although the differences with observations remain relatively small, ranging between 0 - 4 μM south of 30ºS and between 0-2 μM north of 30ºS (Figs. A1a, A1c). Adjacent to the coast between 25º-35ºS, the model overestimates concentrations by ~7 μM (Fig. A1c). In general, differences were minimal over most of the study area, and it can therefore be considered that the model fairly well represents surface $NO_3^-$.

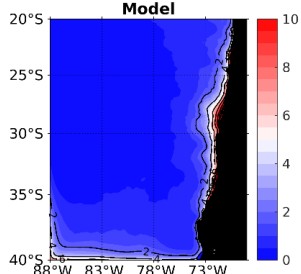 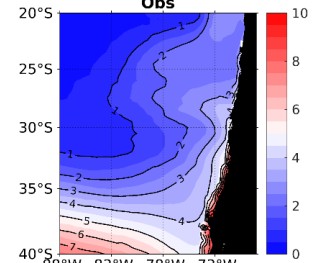 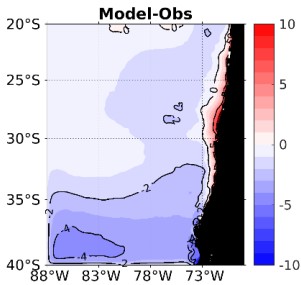

**Figure A1. Spatial distribution of mean surface $NO_3^-$. (a) ROMS-BioEBUS simulation, (b) CARS climatology, and (c) BIAS (model - observations). Units are μM.**

**A.3 Oxygen and nitrate on the isopycnal layers**

The comparison between observations and model is here done over different density layers $S_{upper}$, $S_{lower}$ and $S_{core}$ (Figures A2, A3, A4). The model captures the principal characteristics of $O_2$ and $NO_3^-$, displaying a similar structure and latitudinal and zonal gradients. Compared with observed data, lower $O_2$ concentrations and higher $NO_3^-$ concentrations were predicted near the coast and north of 30ºS associated with the meridional changes of the offshore extension of the OMZ.

In the $S_{upper}$ surface, observations reveal discontinuous areas with lower $O_2$ concentrations north of 25ºS ($O_2$
<100 μM), between 30 - 31ºS and 35 - 36ºS ($O_2$ <150 μM). The model mostly underestimated these $O_2$
concentrations by 40 μM north of 30ºS, by around 20 μM between 30 - 35ºS, and overestimated them by 20
μM between 35 - 38ºS near the slope, but performed better south of 35º S offshore. For $NO_3^-$, observations
indicated very low concentrations mainly along the coast between 23 - 35ºS (<10 μM), with an increase
to >15 μM in the oceanic region. Therefore, the model overestimated concentrations in the coastal zone (>10
μM) and in the northern oceanic zone (4 - 8 μM, Figures A2).
Regarding the $S_{core}$ layer, the model performed better in the northern region, underestimating $O_2$
concentrations between 0 - 10 μM. Off the coast, the bias ranged from -10 to 10 μM. However, south of 30º
S, between 75 - 80ºW, the model overestimated $O_2$ concentrations by over 40 μM, whereas west of 82ºW, the
bias varied between -10 to 10 μM. Nitrate concentration is overestimated by the model in the coastal region
with concentrations exceeding 6 - 8 μM, particularly in the northern oceanic area (2 - 4 μM). The model is
more realistic in terms of $NO_3^-$ in the region west of 78ºW and south of 30ºS (Figure A3).

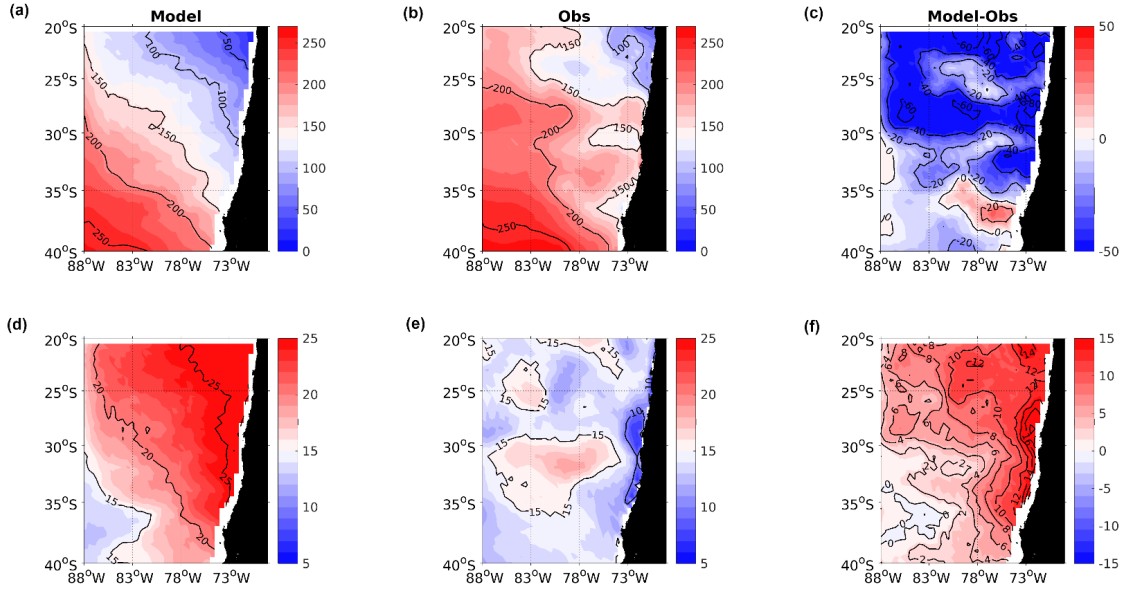


**Figure A2. (top panels, a-c) Mean $O_2$ and (bottom panels, d - f) mean $NO_3^-$ concentration in the isopycnal layer**
**$S_{upper}$ = 26.0 kg m⁻³ from the ROMS-BioEBUS simulation (left), the CARS climatology (middle), and the BIAS**
**(right). Units are μM..**

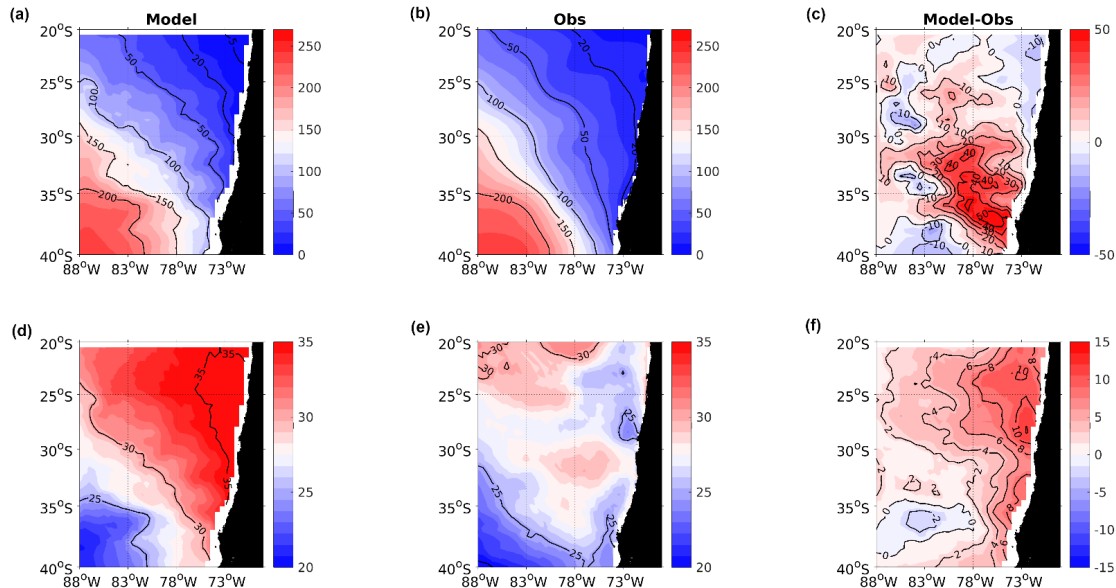

**Figure A3. Same as Figure A2 but for the isopycnal layer $S_{core}$ = 26.6 kg m⁻³.**

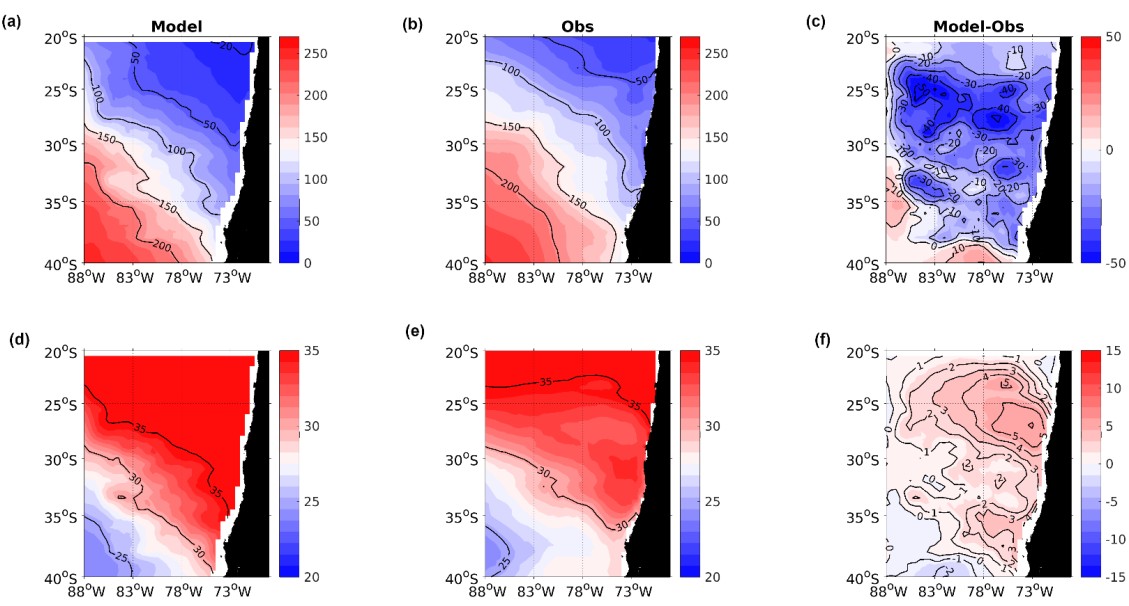

**Figure A4. Same as Figure A2 but for the isopycnal layer $S_{lower}$ = 26.9 kg m⁻³.**
In the $S_{lower}$ layer, the model simulates in general lower $O_2$ concentrations than observations north of 30ºS ,
underestimating by between 10 - 40 μM. This bias is smaller south of 30º S (10 - 30 μM) and in the
southwest (0 - 10 μM). The model realistically simulates $NO_3^-$, with deviations from observations ranging
from -2 to 5 μM and with the greatest overestimation occurring in the region where $O_2 < 50$ μM (Figure A4).

**A.4 Vertical Structure**
The comparison between model and observations is done along zonal sections at 25ºS, 30ºS, and 35ºS
(Figures A5, A6). Along the latitudinal gradient, the contour of 50 μM shows that the OMZ narrows towards
the south. Observations evidence an elongated low-oxygen tongue in subsurface waters that is more
pronounced than in the model simulations, along with an intrusion of oxygenated waters both at the surface
and at depth (~600 m), enclosing the low-oxygen water with a marked oxycline (Figure A5). The oxyclines
showed a greater bias, with the model tending to underestimate the observed data, particularly in the lower
oxycline with an underestimation of up to 60 μM. In the subsurface layer between 200 - 400 m, the model
better represents $O_2$ levels, especially at 25 - 30ºS where differences ranged from 0 - 10 μM (Figure A5c,
A5f). At 35ºS, there is an overestimation (40 μM) between 75-80ºW low 200 - 300 m depth (Figure A5i).

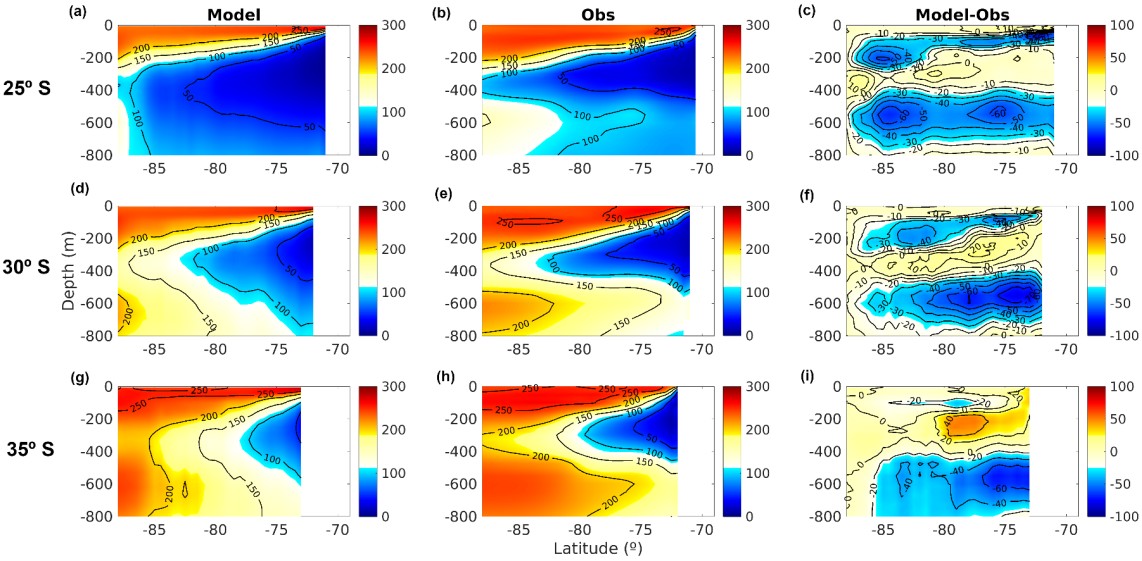


**Figure A5. Zonal section of mean $O_2$ concentration at (a - c) 25ºS, (d - e) 30ºS, and (f - h) 35ºS. (a, d, g) from the
ROMS-BioEBUS simulation (left boxes), CARS climatology (middle boxes), and BIAS (right boxes). Units are
μM.**

Along 25ºS the model simulates higher subsurface $NO_3^-$ concentrations near the coast compared to
observations, showing differences of up to 15 μM. Below 400 m of depth, overestimations range from 0 - 4
μM (Figure A6a, A6b, A6c). These differences decrease towards the south; at 30ºS and 35ºS, the most
significant differences extend from the coast to 76ºW and below -300 m (overestimations of 8 - 12 μM),
whereas elsewhere, the differences remain between 0 - 4 μM. Subsurface $NO_3^-$ is underestimated by 2 μM by
the model at 35ºW and west of 80ºW (Figures A6f, A6i).
Taylor diagrams synthesize the agreement between model and observations for the annual mean $O_2$ and $NO_3^-$
in each of the subregions integrated in the upper 800 m of the water column (Figures A7, A8). Estimates of
standard deviation, correlation, and RMSE reveal that the current model configuration is generally in
reasonable agreement with observations, primarily in the subsurface layer where most of the properties of
Puddies were computed (Figure A9). Therefore, there is confidence in using the model for the
biogeochemical characterization of these mesoscale processes.

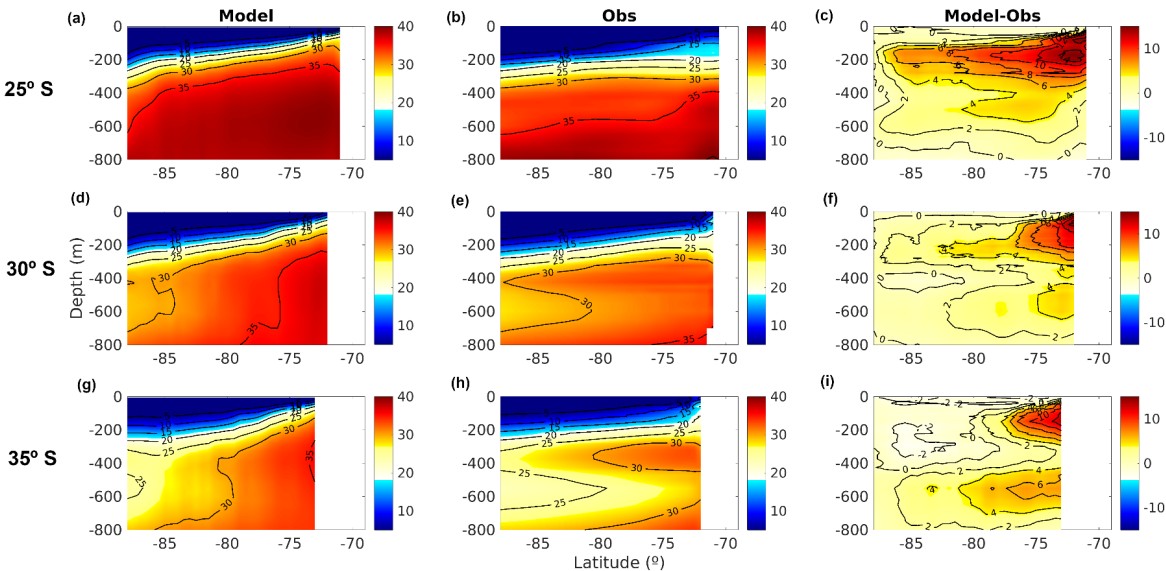


**Figure A6. Same as Figure A5 but for mean NO₃⁻ concentration.**

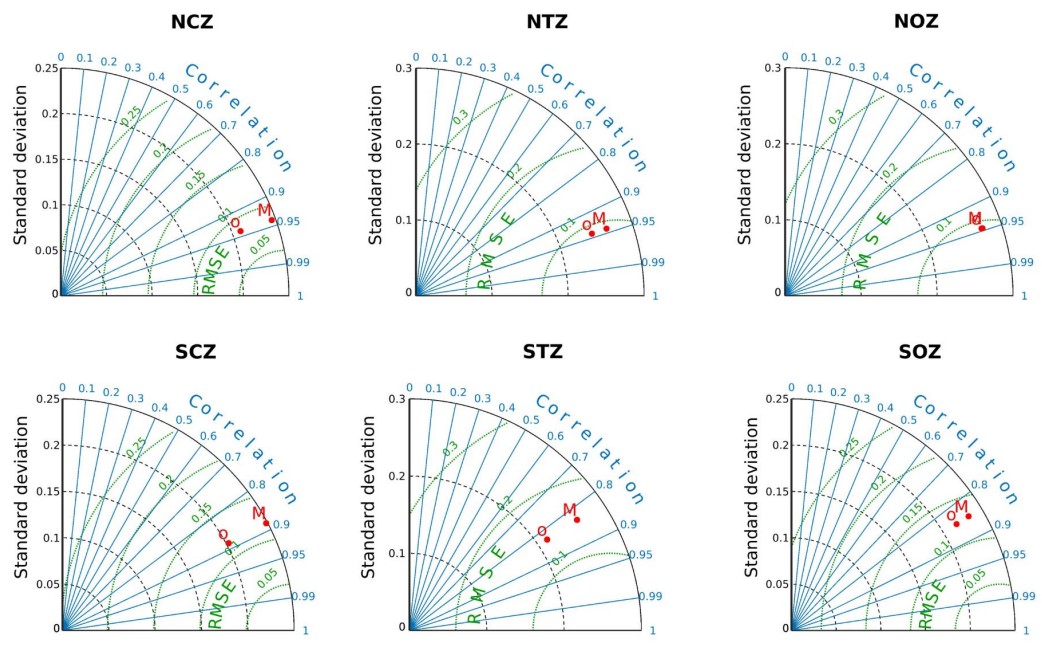


**Figure A7. Taylor diagram for mean O₂ concentrations over the six subregions (Figure 1a) and within the first**
**800m. M is for the model while O is for the CARS climatology. The X-Y axis refers to the standard deviation**
**(black lines), correlations in the radial axis (blue lines) and RMSE is indicated by curved lines (green). Region's**
**names are: Northern Coastal Zone (NCZ); Northern Transition Zone (NTZ); Northern Oceanic Zone (NOZ);**
**Southern Oceanic Zone (SOZ); Southern Transition Zone (STZ) and Southern Coastal Zone (SCZ).**

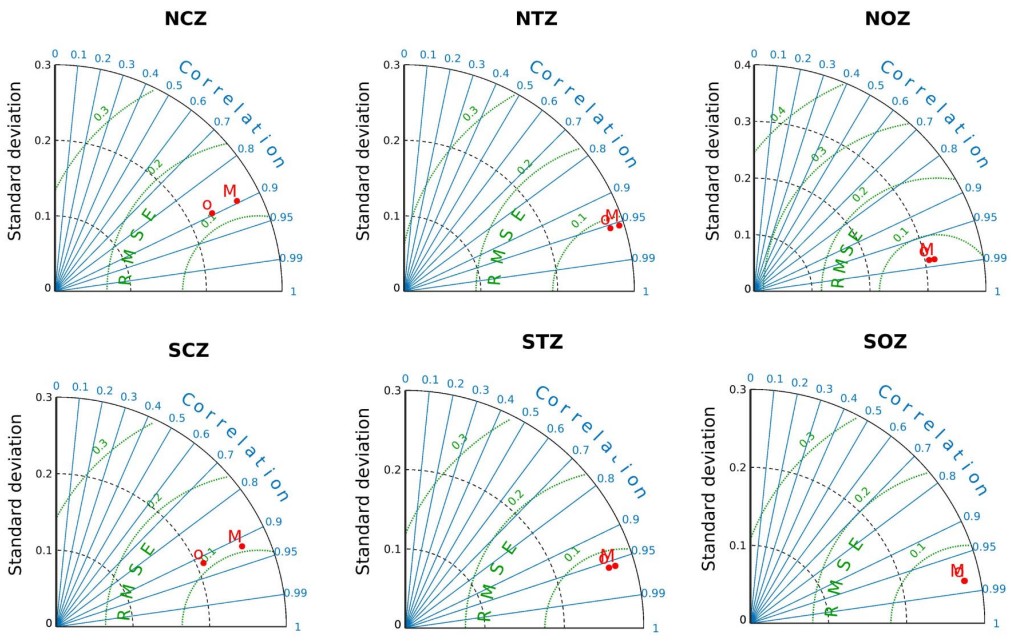


**Figure A8. Same as figure A7 but for mean NO₃⁻ concentration.**

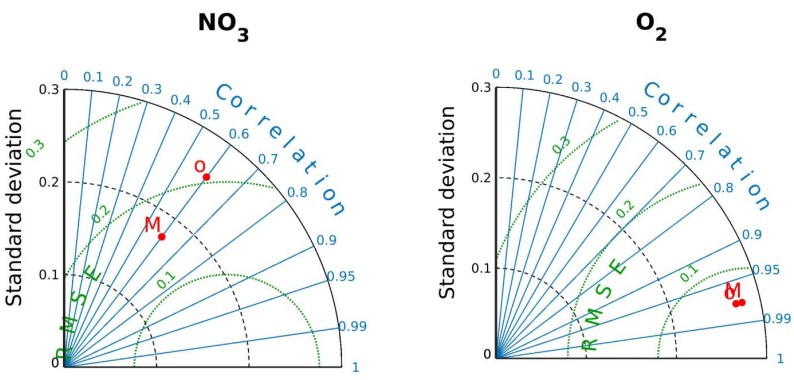


**Figure A9. Taylor diagram for mean NO₃⁻ and O₂ concentrations over the entire S$_{core}$ volume. Details are similar to those in Figures A7 and A8.**

**Code availability**

ROMS model code is available at http://www.croco-ocean.org. All input data set and configuration of our ROMS/BIOEBUS simulations are described in section and in the references therein. This work was granted access to the HPC resources of CALMIP supercomputing center at the Toulouse University under the allocations 2023-1044 and 2024-1044.

**Author contributions**

LOC, OP, MC and BD designed the study. BD performed the simulation. LOC made the assessment, eddy
identification and statistical analysis of the model outputs. OP supervised the project. LOC interpreted the
results and wrote the manuscript with contributions from all co-authors.
**Competing interests**
The authors declare that they have no conflict of interest.
**Acknowledgements**: We thank Dr. M. Pizarro-Koch for useful discussions and Dr. Y. Frenger for her help
with the Faghmous method. LOC was supported by ANID doctoral scholarships and the R20F0002
(PATSER) ANID program. This research was partially supported by ANID (FONDECYT projects: 1181872
& 1241203). OP thanks support from (Millenium Institute of Oceanography grant IC-120019) and Université
of Toulouse Paul Sabatier for funding his stay at LEGOS and CECI during part of the second semester 2024.
BD acknowledges support from ANID (Concurso de Fortalecimiento al Desarrollo Científico de Centros
Regionales 2020-R20F0008-CEAZA, Anillo Eclipse ACT210071 and Centro de Investigación
Oceanográfica en el Pacífico Sur-Oriental COPAS COASTAL FB210021, Fondecyt Regular N°1190276).

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
