# Peer review of "Evolution of biogeochemical Properties Inside Poleward 1"

_EGUsphere, 2024_

## Author Comment (AC2)

**Supplementary material**

**Figures**

Figure S1. Bathymetry of the study zone. The study area extends between 20 - 40°S, from the coast to 90°W and includes the first 1000 m of depth, where eddies have a higher incidence. The color map distinguishes between the marine area (blue and purple colors) and the terrestrial area (brown and green colors). Model configuration includes n=37 sigma levels, with n=28 spanning from the surface to approximately 1000 m depth and 9 additional levels for depths >1000 m. At 30°S we illustrate some sigma levels (black lines) corresponding to the stratification in January 2002 as an example. The displayed sigma level are: n=8 (range: -27 to -47 m), 18 (range: -150 to -260 m), 23 (range: -340 to -570 m), and 28 (range: -760 to -1280 m) These depth ranges are independent of the slope area.

---

## Author Comment (AC4)

**Supplementary material**

**Figures**

[Figure]

**Figure S1. Bathymetry of the study zone. The study area extends between 20 - 40ºS, from the coast to 90ºW and includes the first 1000 m of depth, where eddies have a higher incidence. The color map distinguishes between the marine area (blue and purple colors) and the terrestrial area (brown and green colors). Model configuration includes n=37 sigma levels, with n=28 spanning from the surface to approximately 1000 m depth and 9 additional levels for depths >1000 m. At 30ºS we illustrate some sigma levels (black lines) corresponding to the stratification in January 2002 as an example. The displayed sigma level are: n=8 (range: -27 to -47 m), 18 (range: -150 to -260 m), 23 (range: -340 to -570 m), and 28 (range: -760 to -1280 m) These depth ranges are independent of the slope area.**

[Figure]

**Figure S2. Identification and vertical characterization of a Puddy. It was identify with the Faghmous algorithm where we used the maximum positive anomaly of the layer thickness ($\delta h$, colormap) that is bounded by the density surfaces $S_{upper} = 26.0$ kg m$^{-3}$ and $S_{lower} = 26.9$ kg m$^{-3}$, $\delta h > 0$ indicates the anticyclonic polarity, the largest closed contour around the geographical location of $\delta h_{max}$ is considered as the edge of the eddy (black close contour), as $\delta h_{max}$ is associated with the center of the eddy (black points; see Section 2.5). Eddy was located at 76.5ºW, 35.8ºS with ~21 weeks of life. (b-e) Vertical structure of Puddy shows the isopycnal layers $S_{upper}$, $S_{core}$ and $S_{lower}$ (black contours). The edge (black dotted line) with a radius of 73 km. b) Relative vorticity and velocity field, zonal component velocity (white lines) indicates westward flow (dashed line) and eastward flow (solid line). c) NO$_3^-$, d) NO$_2^-$ and e) N$_2$O. The core with low oxygen (magenta lines) encloses hypoxia (O$_2$ < 45 µM, solid line) and suboxia (O$_2$ < 20 µM, dashed line). White contours enclose oxygen concentrations of 90 and 120 uM.**

[Figure]

**Figure S3. Comparison of the average vertical profile considering an eddy delineated with a circular boundary (orange line) versus an irregular boundary obtained by the Faghmous method (blue line) for each biogeochemical tracer (N₂O, NH₄, NO₃, NO₂, O₂ in µmol) and absolute salinity (in g/kg). The selected times correspond to the early stage (t=10, ~2 months), mid-stage (t=30, ~6 months), and late stage (t=55, ~11.5 months) of the eddy's life cycle. The vertical profile was obtained following the methodology outlined in Section 2.6.2.**

[Figure]

**Figure S4. Mean state of different properties at the first ~100 km from the slope where there are higher occurrence Puddies (strip in color red, Figure 1 - right box). The typical characteristics were obtained averaging in areas of 1ºx1º along the coast between the isopycnals layers S_upper and S_lower by 9 years (See Methods 2.2). (a) absolute salinity, (b) oxygen, (c) nitrous oxide, (d) ammonium, (e) nitrite, (f) nitrate.**

[Figure]

**Figure S5. Mean state of biogeochemical characteristics in the $S_{core}$ layer (26.6 kg m$^{-3}$) in the six subregions. Relations between AOU and (b) absolute salinity, (c) $\Delta NO_3^-$, and (d) $\Delta N_2O$. The subregions are: Northern Coastal Zone (NCZ - dark blue); Northern Transition Zone (NTZ - yellow); Northern Oceanic Zone (NOZ - green); Southern Oceanic Zone (SOZ - magenta); Southern Transition Zone (STZ - brown) and Southern Coastal Zone (SCZ - light blue).**

[Figure]

**Figure S6. Mean vertical variance, and skewness associated with the terms involved in the oxygen budget inside the Puddies at first ~100 km from the slope where there are higher occurrence Puddies (strip in color red, Figure 1 - right box). (a) XADV, (b) YADV, (c) VADV, (d) HMIX, (e) VMIX, (f) PHYS. Details of the terms are similar to Figure 7. The variance (red line) and skewness (blue line) were calculated for each profile following the same method as Figure 3 and averaging along the coast.**

**Tables**

**Table S1.** Average values of AOU (µM), absolute salinity (g/kg), $NH_4^+$ (µM), $\Delta NO_3^-$ (µM), and $\Delta N_2O$ (nM) inside and outside the Puddies (mean state) on the $S_{core}$ surface ($\sigma_\theta$ = 26.6 kg m$^{-3}$). $<\,>_{9-year}$ indicate the mean value of the corresponding variable in the corresponding region, while $<\,'\,>$ the average value of the variable inside the Puddies (see Methods 2.3 and 2.6.1).

| Zones | $< AOU >_{9-year}$ | $< AOU' >$ | $< S >_{9-year}$ | $< S' >$ | $< NH_4 >_{9-year}$ | $< NH_4' >$ | $< \Delta NO_3 >_{9-year}$ | $< \Delta NO_3' >$ | $< \Delta N_2O >_{9-year}$ | $< \Delta N_2O' >$ |
|---|---|---|---|---|---|---|---|---|---|---|
| NCZ | 259.8±1 | 265.0±2 | 34.9±0.01 | 34.9±0.01 | 0.06±0.007 | 0.11±0.026 | 17.85±0.1 | 18.3±0.7 | 28.79±1 | 33.4±2 |
| NTZ | 241.4±2 | 256.6±5 | 34.8±0.01 | 34.8±0.01 | 0.02±0.003 | 0.07±0.03 | 17.1±0.1 | 18.7±0.5 | 23.5±1 | 34.9±3 |
| NOZ | 202.0±22 | 223.9±23 | 34.5±0.06 | 34.7±0.07 | 0.01±0.0003 | 0.017±0.02 | 14.7±1.5 | 16.6±1.7 | 16.4±7 | 26.8±6 |
| SOZ | 124.2±40 | 180.5±31 | 34.2±0.08 | 34.5±0.08 | 0.01±0.001 | 0.02±0.01 | 9.5±2.9 | 13.6±2.3 | 5.4±7 | 17.0±8 |
| STZ | 208.1±3 | 236.8±15 | 34.6±0.01 | 34.6±0.009 | 0.02±0.003 | 0.06±0.03 | 15.4±0.2 | 17.6±1.2 | 24.5±1 | 33.8±6 |
| SCZ | 240.4±1 | 249.8±7 | 34.8±0.004 | 34.8±0.004 | 0.08±0.005 | 0.09±0.03 | 17.7±0.1 | 18.4±1 | 34.65±0.5 | 37.8±3 |

**Table S2.** Percentiles 50$^{th}$, 75$^{th}$, and 90$^{th}$ (P50, P75, P90) of AOU (µM), absolute salinity (g/kg), $NH_4^+$ (µM), $\Delta NO_3^-$ (µM), and $\Delta N_2O$ (nM) on the $S_{core}$ surface ($\sigma_\theta$ = 26.6 kg m$^{-3}$). The profiles were obtained from random samples (# pro.) using Montecarlo's method, for each region.

| Reg | # pro. | AOU | | | S | | | $NH_4^+$ | | | $\Delta NO_3^-$ | | | $\Delta N_2O$ | | |
|---|---|---|---|---|---|---|---|---|---|---|---|---|---|---|---|---|
| | | P50 | P75 | P90 | P50 | P75 | P90 | P50 | P75 | P90 | P50 | P75 | P90 | P50 | P75 | P90 |
| NCZ | 1246 | 257 | 266 | 269 | 34.72 | 34.77 | 34.79 | 0.06 | 0.12 | 0.17 | 18.08 | 18.47 | 18.87 | 32.48 | 35.6 | 39.56 |
| NTZ | 1474 | 244 | 254 | 261 | 34.6 | 34.64 | 34.69 | 0,009 | 0,019 | 0,048 | 16.98 | 17.92 | 18.61 | 25.28 | 32.24 | 37.78 |
| NOZ | 3509 | 209 | 224 | 239 | 34.48 | 34.53 | 34.58 | 0,002 | 0,004 | 0,008 | 14.38 | 15.84 | 17.2 | 15.7 | 23.03 | 29.66 |
| SOZ | 6808 | 72 | 95 | 121 | 34.12 | 34.16 | 34.22 | 0,006 | 0,009 | 0,013 | 5.58 | 7.74 | 9.60 | -1.44 | -1.11 | 1.77 |
| STZ | 521 | 222 | 236 | 248 | 34.49 | 34.55 | 34.6 | 0,018 | 0,033 | 0,054 | 16.36 | 17.52 | 18.52 | 29.07 | 34.96 | 41.13 |
| SCZ | 235 | 240 | 256 | 264 | 34.57 | 34.65 | 34.7 | 0,052 | 0,119 | 0,188 | 17.69 | 18.69 | 19.44 | 36.14 | 42.96 | 47.11 |

---

## Author Response (AR1)

**Anonymous Referee #1, 11 Sep 2024**

First, let me apologize for the delay in providing my comment for the manuscript, which greatly affects the review process. Please accept my sincere apology.

In this manuscript, by employing a high-resolution coupled simulation of the Southeast Pacific, Ortiz and colleagues characterized the variabilities in internal biogeochemistry, disparities with external properties, and processes influencing the dissolved oxygen (O2) budget of Poleward Undercurrent Eddies (PUDDIES) and scrutinized eddy dynamics and biogeochemical processes associated with the nitrogen cycle in those PUDDIES during their transit to oceanic waters. Specifically, they documented the evolution of water mass properties and processes inside PUDDIES with contrasting initial characteristics (suboxic versus hypoxic) in order to evaluate the role of these properties in the maintenance of oxygen minimum zones (OMZs). They found that Puddies capture a biogeochemical signal contingent upon their formation location, particularly associated with the core of the Peru-Chile Undercurrent at the core of the OMZs. Furthermore, they found that while permeability at the periphery facilitates exchange with external waters, the core signal retains negative O2 anomalies and positive anomalies of other biogeochemical tracers. These disturbances contribute significantly to average properties that exceed background conditions in the open ocean. The main process controlling O2 input into, or output from the eddy core entails lateral and vertical advection, with vertical mixing supplying O2 to a lesser degree. Biological activity consumes O2 inside PUDDIES for around 6 to 12 months, especially intensely for the first 100 days, thus facilitating the persistence of low O2 conditions and extending the lifetime of biogeochemical anomalies within the core. The authors suggested that by representing more complex biogeochemical and physical processes in ocean models, future studies will be able to quantify the effects of other SMS processes within Puddies that are not yet considered in their study. Validation and supplementary statistics of Puddies through further observations are urgently needed.

The manuscript is well written and the method and model setup are well-designed. The authors also did a thorough job on the transformations and transport of biogeochemical elements inside PUDDIES. I have learned a lot and have enjoyed reading the manuscript. My only reservation regarding the manuscript is that I feel the authors can do a better job on producing Figures. Some figures (especially figure 1) are small with a bad color scheme that makes it difficult for readers to see and interpret the (sometimes) dense information that the authors try to convey.

We appreciate all the suggestions and the constructive comments. Following the recommendations, we have improved the presentation of the figures.

Other detailed comments.

Line 17: Please change to: "Oceanic eddies are ubiquitous features of the circulation thought to be involved…"

Done

Lines 30 - 31: "These disturbances likely contribute to average properties that exceed the 90th percentile threshold in the open ocean, contrasting with the formation zone where they surpass 50th percentile levels." Please consider adding a clear definition of the percentile threshold levels here.

The statement was clarified as follows:

"These anomalous conditions result in tracer values exceeding the 90th percentile of their distribution in the open ocean, in contrast to the formation zones, where anomalies only surpass the 50th percentile. This indicates that Puddies may play a role in modulating the average properties of the open ocean"

Line 37: What do you mean by "out of time"?

It was changed: earlier

Figure 1: Please consider improving the quality of Figure 1: Bigger size, bigger font for texts, different color scale other than the rainbow, clearly distinguish between black lines and black contour, clearly show the yellow dots tracking the Puddies… They are all very hard to see now.

Thanks for your suggestion. Figure 1 was modified following the recommendations and it now only includes two panels allowing for a bigger size and font.

Line 189-190: Please change to "which is a scalar"

Done

Line 201: Is the Score defined as density core? Please clarify (I know you defined it later but it should be defined clearly earlier).

We changed the text to clarify: "is centered at a density surface of $\sigma_\theta$ = 26.6 kg m$^{-3}$ ($S_{core}$)"

Line 237: Please change to N2O production

Done

Line 369-370: Please change "The correlation coefficients varied between 112.5 and 457.45" to "The slope values…"

The text was changed: "The slope values ranged from 112.5 to 380 for AOU and 142 to 412 for O$_2$ fit"

Line 490 and Line 493: please add < 0 or > 0 delta O2/delta z here.

That part was removed.

Table 6: Probably a naive question, and please excuse my ignorance, but can you clarify what are x and y used here to calculate the decay rate k?

We have added a brief definition of the model in the description (Table 6 changed to Table 4): " The decay rate derived from an exponential fit of the form $y = Ae^{-kx}$, where $x$ represents the distance from the coast in km and $y$ is the concentration of the biogeochemical tracer. $k$ is the decay rate and $A$ the tracer concentration in the coastal strip (1°x1°). A zonal extent of 1,445 km was used at 25°S and 30°S, while at 36°S we used 1,334 km (i.e. same range of longitude as in Figure 8). Values of k are statistically significant at the 95% confidence level".

Line 614: Please change to: "affected also by the southward reduction in contribution of ESSW"

The text was changed to: "also affected by the southward reduction in the contribution of ESSW"

Line 624: Please replace Table XX here with the correct table.

We have added a new figure with the percentiles information (new Figure 5) and the Table was moved to the Supplementary material (Table S2).

Line 690: This is the first instance in the text that I see the acronym CARS. Please define it.

"CARS" stands for "CSIRO Atlas of Regional Seas" but was changed to "climatological observations" for clarity.

**Anonymous Referee #2, 01 Nov 2024**

Review of Ortiz et al. "Evolution of biogeochemical Properties Inside Poleward Undercurrent Eddies in the Southeast Pacific Ocean"
The paper by Ortiz et al. aims to characterize the biogeochemical evolution of poleward undercurrent eddies (PUDDIES) generated in the Southeast Pacific OMZ. To do so, the authors use 9 years (2000-2008) of output of a physical-biogeochemical coupled model (ROMS+BioEBUS) run on mesoscale resolving grid (1/12 degree resolution ~8km) and saved as 3-day means. The authors distinguish between the evolution of PUDDIES characterized by initial hypoxic conditions and those with initial suboxic conditions, contrast the evolution of water mass properties inside and outside of the eddies and focus on a few case studies to determine changes in the eddy properties along their trajectories. The authors use Reynolds decomposition and eddy tracking methods to study the impact of the PUDDIES on the oxygen and nutrient fields.

General comment
The topic and idea of the paper are interesting, but I'm not convinced by many of the choices made by the authors and I find the paper difficult to read and its results difficult to interpret. I have decided not to go into the details of the results for this reason, but to comment on the methods at this stage. I think this manuscript and its related research needs major revisions

before being published. I'll be happy to provide a more detailed review on the results once the methods are better justified.

We thank the reviewer for his/her constructive feedback. Below, we provide point-by-point responses to each comment.

Many of the choices in the methods need to be clarified. The domains in which the region of study is split (8 domains) seem odd and not well justified in the methods (especially the latitudinal boundary, which seems arbitrary). Some of these domains are very narrow and coastal (H) where the authors state the algorithm has trouble identifying eddies (lines 299-307), some mix coastal and offshore zones (G,F), some are very wide (E). All have irregular shape and mix zones at a variety distances from the coast (where eddies are generated). This makes for a statistical and geographical nightmare in understanding and interpreting the data. This is a major problem for me, because all the results are presented in terms of averages across these oddly shaped and dishomogeneous regions.

Following the reviewer's recommendation we have clarified the method for selecting the domains. It should be first recalled that selected regions have an irregular shape because they depend on mean conditions that somehow depend on the horizontal distribution of the oxygen minimum zone delimited by the dominance of ESSW. That said, for clarity, we have reduced the number of biogeochemical zones or study areas and the method has been expanded (see section 2.2). The new figure 1a displays the limits of the new domains and we present in the new figure 2 the biogeochemical contribution of the different water masses in each area. The new areas were defined to encompass a range of similar biogeochemical characteristics. Given the importance of the ESSW, as it is the water mass primarily enclosed in the Puddies, the isopycnal surface associated with the ESSW and the OMZ core (isopycnal layer 26.6 kg m$^{-3}$) was used as a reference, similar to the approach used by Pizarro-Koch et al. (2019) for this model.

The authors interpolate 37 vertical layers of model output to a regular grid of 160 vertical layers generating a lot of "fake data" for their analysis.

Section 2.2 and Figure S1 clarify the number of sigma levels contained in the first 1000 m. The vertical resolution of 5 meters to infer tracers on a regular vertical grid through linear interpolation is commonly used (e.g. Illig et al. (2018)), but it was checked that the results are a little sensitive to the interpolation method (spline versus linear) or a number of z-levels retained (linear interpolation each 5 m, 10 m and 20 m to 160 z-levels were tested). For consistency with previous studies, we used a vertical resolution of 5 meters.

SENSITIVITY ANALYSIS USING VERTICAL INTERPOLATION WITH DZ=5M, 10M, AND 20M

[Figure]

Figure 1. Comparison of isopycnals 26, 26.6, and 26.9 kg/m³ using different vertical interpolations (dz: 5 m, 10 m, and 20 m) for the modeled average oxygen (9 years). The contours (black lines) represent oxygen concentrations of 50, 100, 150, 200, and 250 μM.

[Figure]

Figure 2. Comparison of isopycnals 26, 26.6, and 26.9 kg/m³ using different vertical interpolations (dz: 5 m, 10 m, and 20 m) for the modeled average nitrate (9 years). The contours (black lines) represent nitrate concentrations of 25, 30, and 35 µM.

Vertical interpolation with dz=5 m allows us to reduce the variability of the isopycnal surface, particularly for the isopycnal $S_{upper}$=26.0 kg/m³. Since this study focuses on analyzing mesoscale processes, it is important for the identification and analysis of 3D eddies to minimize smaller-scale variability, such as that observed with coarser interpolations (dz = 10 m and dz = 20 m).

The choice of "reference average" for the Reynolds decomposition is unclear.

The sentence was modified in the text for clarity: "where $< NO_3 >_{9-year}(x, y, z)$ is the "mean state" of $NO_3^-$ calculated as an overall mean across 9 years simulated between the period 2000-2008. Fluctuations of this "mean state" thus consider the large intraseasonal and seasonal variability and are denote as $NO_3'(x, y, z, t)$."

The authors say that they use a circular eddy mask to extract the eddy properties, when the Faghmous algorithm provides the actual shape of the eddy contous, which provides a more accurate definition of the eddy core properties. A lot of these choices need to be better justified and a few probably should be revised.

A sensitivity analysis was conducted to determine the optimal parameterization of the Puddies in 3D, comparing a regular circular edge to the irregular edge generated by the Faghmous algorithm (Figure S3). The results indicated that the irregular edge provided by the Faghmous algorithm was more effective. Consequently, we recalculated the sections involving 3D Puddies using this criterion. The methodology is detailed in Section 2.6.2.

The authors also use 9 years of model output at 3-day mean resolution, but then only use two eddy tracks (among likely tens or maybe hundreds found by the employed algorithm) as "case studies". There is no proof that the two tracks are representative of the eddy populations. Why not building an average of the eddies in time along their tracks so that eddies of the same age are averaged together? This was successfully done in previous studies (https://doi.org/10.5194/os-15-1111-2019 ) and it would allow to use a statistical approach that really takes advantage of the large amount of information provided by the model data. The model data available for the study is so abundant that the current approach seems reductive.

Following the reviewer's recommendation, we conducted a more robust analysis by considering all Puddies identified at each time step (t = 3 days) over a 9-year period across the three selected latitudes. This approach enhances the statistical significance of the results, providing deeper insights into how the internal properties of the eddies evolve as they move away from the coast and the extent to which these perturbations influence the mean state. Additionally, this analysis allowed for a clearer characterization of the differences between the selected latitudes. The methodology is detailed in Section 2.6.2,

and the results are presented in Section 3.4 (Subsurface water band changes caused by Puddies). Figures 8–11 and Table 4 summarize the findings.

I have brought up many of these concerns also in the detailed comments.
The paper is also difficult to read due to the Methods section being poorly organised and missing important detail and the figures not having appropriate labelling, so the reader has to note what the figures represent by hand (impossible to do if reading the paper online). The Supplement is currently unreadable, see more comment below.

We appreciate your detailed comments and constructive feedback. We understand your concerns regarding the organization of the Methods section and the lack of clarity in figure labeling, as well as the readability issues with the Supplement.

To address these points:

Methods Section: We have restructured the Methods section to improve its clarity and organization. Additional details have been included to ensure that the methodology is thoroughly explained and easier to follow.

Figures: All figures have been revised to include appropriate and detailed labeling, making them fully self-contained and accessible for readers. Figure captions have also been expanded to ensure clarity.

Supplementary Material: We have reviewed and reformatted the Supplementary Material to ensure it is clear and readable. Specific changes include [briefly mention any significant changes made, e.g., reorganizing content, adding labels, or improving formatting].

We hope these revisions adequately address your concerns and improve the overall readability and quality of the manuscript. We are grateful for your valuable input, which has been instrumental in refining the paper.

Detailed comments

1. Line 49: "Under these conditions, heterotrophic metabolic processes prevail" - what conditions? Prevalence of heterotrophic processes is mostly due to being at depth and hence below the sun-lit layer.

For more clarity, lines 49-52 were modified as follows: Under these conditions of low oxygen in the subsurface (i.e. dissolved $O_2$ less than 20 µM), heterotrophic metabolic processes are important, dominated by activity of bacteria and archaea, resulting in significant shifts in biogeochemical cycles (Lam et al., 2009; Paulmier & Ruiz-Pino, 2009; Wright et al., 2012).

2. Line 60: "where anoxic conditions can even be observed" - this piece of sentence need revising, it doesn't read correctly in English

The text was changed to: "where even anoxic conditions can be observed"

3. Line 67: "through the global warming" - please remove "the"
Done

4. Line 72: "turbulent dynamics" - I'd be careful here using the word "turbulent" since turbulence is a very broad term that in physical oceanography describes a large variety of scales of physical processes down to sub-meter scales. I would rather use "meso- and submesoscale" instead, which is more in line with the topic of the paper. Some processes such as "turbulent mixing" are most likely parameterised in models, but this isn't what the authors refer to.

Thank you for this comment. The text was changed to "sub to mesoscale dynamics"

5. Lines 72-55: I would add some support to the claim of the importance of mesoscales in the OMZ representation in models, currently unsupported in the paper. Literature suggestion: https://doi.org/10.1029/2022MS003158

Following the reviewer's recommendations, we have added relevant references to highlight the importance of sub to mesoscale dynamics in accurately representing OMZs in models. The added text is as follows: "While in current generation climate models, the changes in mean circulation either in the tropics or the high-latitudes, mediated by diapycnal mixing, are invoked to explain the ventilation process (Pitcher et al., 2021), regional modeling studies highlight the importance of mesoscale dynamics, such as mesoscale eddies and zonal jets, in expanding and ventilating oceanic zones with oxygen deficits (Bettencourt et al., 2015; Auger et al., 2021; Calil, 2023) ".

6. Lines 117-118: Unclear sentence

The sentence was modified as follows: "We analyze the physical and biogeochemical factors influencing the variability in the lifespan of elements trapped within the Puddies, with a particular focus on those related to the nitrogen cycle, as they travel across the OMZ towards better-ventilated oceanic waters"

7. Line 196: What about zooplankton respiration?

Zooplankton respiration was considered together with the excretion in this parametrization.

8. Line 203: The authors interpolate the model output to a regular grid of 5m of vertical spacing between 0-800 m depth. This means interpolating to 160 vertical levels! The model output has only 37 vertical levels, of which 13 are found "in the deep ocean", hence possibly only 24 model grid levels are in the range of interest. Why generating so much artificial data?

As previously mentioned, this approach is commonly used with model outputs on a sigma-level grid to determine the specific depths of isopycnals or isotherms. Estimating the error associated with the choice of the number of z-layers or interpolation methods is challenging, as it would require a thorough investigation into the model's sensitivity to vertical resolution. Such an analysis, however, falls beyond the scope of the present study.

9. Line 204: "In the deep ocean (~4000 m depth) typically 13 of the 37 vertical levels fall within this depth range." - You need to indicate the specific depth range, ~4000 m is not a range.

Of the 37 vertical levels, 28 are distributed within the upper ~1000 m of the water column, while the remaining 9 levels are allocated to depths greater than 1000 m. (See new Figure S1).

10. Lines 207-212: I don't understand the need to split the domain at 30°S. Is there a reason why this exact latitude is significant?

This was specified in the text as follows: "The latitudinal division was based on the influence of the characteristic water masses in the region, primarily the Subantarctic Water (SAAW; 11.5°C, 33.8) south of 30°S, where the OMZ is predominantly hypoxic ($O_2$ < 45 μM; as described by Naqvi et al., 2010, and Pizarro-Koch et al., 2019) and significantly narrower. The biogeochemistry of SAAW differs notably from that of Subtropical Water (STW; 20°C, 35.2) north of 30°S, which features a zonally broader OMZ characterized by suboxic conditions ($O_2$ < 20 μM, Figure 2; following Wright et al., 2012), particularly in oxygen, ammonium, and nitrite" (Section 2.2).

11. Line 211: "Pudies" - Puddies? Also, the choice of acronym must be homogenized across the manuscript: all capital (PUDDIES) as in the abstract, or capital P only (Puddies) as here?

We have homogenized the text using Puddies.

12. Lines 213-214 and Figure 1c: If the first 100 km are such a relevant range for the formation of Puddies, why not looking at zones that have boundaries at equal distance from the coast, so that the 0-100 km zone is the formation zone across the whole system and then more offshore zones can be defined?

Puddies are indeed formed in the coastal domain along the coast of central Chile, but the subregions are defined based on mean conditions over the whole domain (coast and open-ocean). The criteria used for the division of the subregions are now better explained (Section 2.2).

13. Lines 217-228: What is your definition of mean state? This must be made explicit here. Is it an overall mean across your 9 years of simulation? In this case, seasonal variability will contribute to the "fluctuations" as it will be part of the residual field. Or else, is it something like interpolated seasonal or monthly climatological means? Please specify.

We computed an overall average over the 9 years of simulation to define the "mean state." Certainly, seasonal variability can contribute to fluctuations, potentially increasing the biogeochemical variability in the cores of the Puddies (Figure 4). However, this does not significantly affect the results, as comparisons were made with other reference points, such as different percentiles (Figure 5). A more explicit definition of the mean state was provided in Section 2.3.

The sentence was modified as follows: "where $<NO_3>_{9-year}(x, y, z)$ is the "mean state" of $NO_3^-$ calculated as an overall mean across 9 years simulated between the period 2000-2008. Fluctuations of this "mean state" considered the seasonal variability and denote as $NO_3{}'(x, y, z, t)$.""

14. Line 261: Did the algorithm identify eddies larger than 300 km of diameter, and if so how many? This seems very large and surprising to be identified as eddy for the Faghmous et al. 2015 algorithm run on an 8km grid.

Puddies with a diameter >300 km were not always identified, but in the snapshots where one was identified, it reached a maximum of 3 that were removed from the analyzed sample. It is important to consider that the Faghmous algorithm defines the eddy boundary based on a single maximum (for anticyclones), closing the boundary when another maximum (or minimum) is found (Section 2.5). This causes the boundary to change in shape and size at each time step. Therefore, it is possible that at time *t*, a Puddy may have a diameter greater than 300 km, and at *t+1*, it no longer does, meaning that it would be included in our statistics.

15. Line 262: "every three days" - does this simply refer to the fact that the model output was saved as 3-day means? If so, please rephrase, since a 3-day mean is different from "every three days" output, which might mean a simple snapshot of the model every three days

Thank you for pointing out this inconsistency. The text was clarified: "Eddy identification was performed using 3-day averages across the entire period".

16. Lines 247-264: How many eddies were identified in total by the algorithm over the model output? How many independent tracks did they correspond to? This is important information to be included here.

On average, around 14 Puddies were identified at each time step (it ranged from 11 to 20 eddies), this was mentioned in Section 3.1. For this study, it was not necessary to track the trajectory of each Puddy.

17. Lines 265-271: I'm puzzled by this choice of only analyzing the tracks of only 2 Puddies when the authors have 9 years of model output available at three days mean output resolution, which most likely allows the authors to analyze tens or maybe hundreds of eddy

tracks. Is there any reason why these two tracks should be particularly relevant or representative of the mean eddy for the northern and southern subregions?

Following the reviewer's recommendation, we conducted a more robust analysis by considering all Puddies identified at each time step (t = 3 days) over a 9-year period across the three selected latitudes (Section 2.6.2 and 3.4). Figures 8–11 and Table 4 summarize the findings.

18. Lines 286-290: Why do you apply a circular mask that will surely miss part of the eddy shape and likely include part of non-eddy waters (eddies are rarely perfect circles), when the Faghmous et al. 2015 algorithm already provides the identified eddy perimeter mask?

For clarity the text was modified as follows: " For these Puddies, we estimated characteristics within the entire eddy volume as follows: i) for each $\delta h_{max}$ (Puddy center), a mask of the entire volume was created; ii) the area mask was calculated to identify the edge using the Faghmous et al. (2015)'s algorithm, which was found to be most accurate approach (Figure S3; see sensitivity analysis in the Supplementary Material); iii) this mask was applied across all depths (at 5-meter intervals), creating an irregular cylinder between $S_{upper}$ and $S_{lower}$; iv) from the total volume enclosed by the cylinder, an average vertical profile was obtained at each time step for each variable (see an example of vertical Puddy characterization in Figures S2b-e)". Details in Section 2.6.2

19. Line 300 – how do you define a "puddy profile"?

A brief definition is now provided: The added text is as follows "To understand the typical conditions within Puddies identified in the formation zone and each subregion, average profiles were constructed as follows: i) all eddy centers (i.e., $\delta h_{max}$ positions) were classified in 1°x1°  grid cells along the coast and also within the regions defined in Figure 1a, ii) for each eddy center, vertical profiles were extracted between the density surfaces $S_{upper}$ and $S_{lower}$ for all variables in each corresponding region, iii) then, these profiles were time-averaged to obtain typical profiles for each variable. For the different regions, the values associated with the $S_{core}$ are shown (Section 3.2) to emphasize changes in the ESSW core. "

20. Lines 304-307: This seems like an important caveat and it needs better clarification. What does "slightly overestimated" mean in numbers? Please provide a number of how many identified Puddies are actually not Puddies but coastal trapped waves or other structures. In other studies, coastal eddies in the first life stages have been excluded from the analysis just because the Faghmous et al. 2015 algorithm was having difficulty identifying them.

Original statement: "It should be noted that the number of eddies in the coastal region may be slightly overestimated by our adopted algorithm due to difficulties in distinguishing between perturbations of the density surfaces generated by eddies, coastal upwelling events, coastally trapped waves or meanders of the coastal currents."

We have revised our statement, as it was previously unclear and partially inaccurate. Coastal waves and upwelling events, in fact, exhibit larger spatial scales than Puddies and would not be misclassified as such based on our criteria. Specifically, only closed-contour structures with a minimum horizontal area of 30 grid points ($A_{min}$ ~1.95 × 10$^9$ m², corresponding to a radius of ~25 km) were retained. This threshold ensures that the identified structures are distinct from coastal-trapped Kelvin waves or upwelling events. In order to provide an error estimate on the number of Puddies, we have applied the Faghmous et al. (2015)'s algorithm varying the criteria on the number of grid points to detect the closed-contour structure. When selecting an area with 25 grid points, the eddies were not properly identified (reducing the total of selected Puddies by 30%). Additionally, eddies with a radius >150 km (~32 pixels in diameter) were also not well detected, with at most three closed-contour structures per snapshot being discarded.

This is now indicated in the main text which was revised as follows: "Only eddies that reached a minimum horizontal area of 30 grid points ($A_{min}$ ~1.95 × 10$^9$ m², equivalent to a radius of ~25 km) were considered, as eddies below this threshold are not well identified (reducing the total of selected Puddies by 30%). Additionally, only eddies with a radius not exceeding 150 km (~32 pixels in diameter) were included, as larger eddies are also not well detected, with at most three closed-contour structures per snapshot being discarded." The methodology was adjusted in Section 2.5 and the original statement was removed in Section 3.1

21. Figure 2: Please add labels to this figure, what does each line represent? It's not enough to have it in the caption, it makes the figure really difficult to understand.

Following the reviewer's recommendation the presentation of figure 2 (now new figure 3) was improved.

22. Table 1: Number of pixels? What does it mean?

Table 1 was modified. The number of pixels refers to the area which is now expressed in km$^2$ for clarity.

23. Tables (in general), Results and Methods: I'm really not convinced by having these 8 different regions. It makes the results overly complicated to understand. I think the authors should rethink this analysis almost entirely and use more regular domains, for example defined by offshore distance bands.

The use of regular domains is not appropriate because of our definition of Puddies based on the thickness between two selected mean isopycnals where the core of the Puddies is expected to be found. We have clarified the text in the method section and have retained only 6 subregions (instead of 8 initially).

24. Tables (in general): These tables are heavy to read. They need to be summarized into figures, bar plots, or something that makes it possible to the reader to grasp the results.

Following the reviewer's recommendation, the presentation of the tables was improved and their number was reduced to 4 (instead of 6). To summarize the results of the two tables, we now have the new figure 5 (the tables are now presented in the Supplementary material).

The supplement is currently unreadable. It's impossible to revise a document where figures and captions are provided separately in different PDFs. I had to copy by hand the text onto a word document and then screenshot the figures and stich them on top of the correct captions in the word document, to be able to understand what I was looking at. The quality of the supplement should be at least checked by the journal upon submission. The supplement should be revised entirely to make it clear to the reader. There are also typos in the supplement's captions and figure axes miss titles and key information.

We apologize for the inconvenience. We have improved the presentation of the Figures and caption text of the Supplementary material. Proof reading was also performed to avoid typos and errors.

**Olivier Sulpis, 07 Nov 2024**

The manuscript combines a comprehensive observational dataset and physical-biogeochemical numerical simulations to analyze the origin of Poleward Undercurrent Eddies (Puddies) and their transports of biogeochemical properties in the Southeast Pacific Ocean. The topic is interesting and relevant. The manuscript is well structured and presents enlightening results about the subsurface mesoscale structures formed in the Peru-Chile Upwelling System. In my opinion, the manuscript can be published after addressing the minor comments listed below.

MAJOR COMMENT

- Some of the most important results are based on the definition of the isopycnal surfaces that bound the OMZ. This should be justified. Although they can be supported by the literature, these values might not be adequate for your model data.

We appreciate the comments and suggestions that have been helpful in this work.

The justification was presented in the text as follows: "The separation of coastal, transitional, and oceanic zones considered the dominance of Equatorial Subsurface Water (ESSW; 12.5°C, 34.9) in the subsurface waters and its increasing oxygenation farther from the coast (Silva et al., 2009). Given the importance of ESSW in defining the study areas and as the primary water source enclosed in the Puddies, the isopycnal surface associated with the ESSW and OMZ core was used as a reference. Specifically, we focused on the isopycnal with a density of 26.6 kg m$^{-3}$, corresponding to the reference OMZ core in this model (Pizarro-Koch et al., 2019), which was used in part of our statistical analysis".

MINOR COMMENTS

- Pg1, L22: PUDDIES, Puddies, use only one convention.

Done

- Pg4, L119: puddies, Puddies.

corrected

- Pg4, L134: Mention the resolution here (by the way, that is not that high).

The model resolution was mentioned. It is 9 km.

- Pg4, L138: were, was

corrected

- Pg4, L144-145: "…resolution of… 37 levels… suitable for resolving mesoscale features", please discuss briefly about this.

Mesoscale features are defined primarily based on their horizontal scale, not their vertical extent. Eddies, in particular, exhibit significant vertical dimensions, making the use of 37 vertical levels suitable for capturing their structure. Additionally, the model operates on a sigma-level grid, with most levels concentrated in the upper ocean, enabling the resolution of relatively small-scale 3D recirculation features effectively.

- Pg4, L147: 20°, 20°S.

corrected

- Figure 1: Dots of the trajectories of the Puddies are too small.

We have modified Figure 1 and improved its presentation. The trajectories are no longer shown.

- Pg5, L154: "Black lines…", where? In panel (b)?

Figure 1 was modified

- Pg5, L157: "Yellow dots", where?

Figure 1b has been removed, and the methodology has been adjusted to address the feedback provided by the other reviewers.

- Pg5, L160: "the possibility that the same eddy…", has you considered using an eddy-tracking method to avoid accounting for the same eddies at the same grid cell?

Faghmous et al. (2015)'s algorithm is an eddy-tracking method but has some issues with the paths of the eddies. Subsurface eddies are difficult to track regardless of the method.

- Pg5, L178-179: "Phytoplankton biomass was estimated…", what about the one from the model?

The estimated phytoplankton biomass is actually from the model. This was clarified in the text.

- Pg6, L195-196: "For O2, the source…", any reference?

The reference is the following and was added in the text: "(Peña et al., 2010)."

The full reference: Pena, M. A., Katsev, S., Oguz, T., and Gilbert, D.: Modeling dissolved oxygen dynamics and hypoxia, Biogeosciences, 7(3), 933-957, https://doi.org/10.5194/bg-7-933-2010, 2010.

- Pg6, L199: "…is centered at a density surface of …", how is this assumption supported?

We clarified this assumption in the text: "following as reference the core of ESSW near coastal zones, which is similar to Pizarro-Koch et al., (2019; see Section 2.2)".

- Pg6, L202: "…the 1000 m and 500 m depth isobaths", where?

The lines 209-210 were changed as follows: "Near the slope, a deepening in $S_{core}$ is observed north of 30°S, where the slope is narrower whereas south of 30°S, it widens."

- Pg6, L211: Pudies, Puddies.

Corrected

- Pg7, L225: "…with annual and interannual variability", and seasonal?

Annual was meant to indicate the seasonal variability, but we agree that this is confusing. For clarity we now use the term seasonal to indicate the annual variability

- Pg7, L229: Define AOU, <Delta>NO3, <Delta>N2O.

The text has been changed to: "The apparent oxygen utilization (AOU), $NO_3^-$ production ($\Delta NO_3^-$), and $N_2O$ production ($\Delta N_2O$) provide an estimate of how much has been produced/consumed by biological processes since the water mass was formed in terms of oxygen, nitrate, and nitrous oxide respectively."

Additionally, the text of Section 2.4 was expanded for clarity.

- Pg7, L235: Substitute "Eq. (3)" by "relation" or "equation".

That's the journal's format for naming equations

- Pg7, L236: Explain what subscripts "sat" and "obs" mean.

Done. The subscript "obs" was removed to avoid confusion, as we used modeled variables for this study.

- Pg7, L237: Sarmiento, Sarmiento's.

corrected

- Pg7, L238:  Delete "Eq. (4)".

The line was changed to: the following relationship, Eq. (4):

- Pg7, L240:  Delete "Eq. (5)".

Done

- Pg8, L253:  How are "S_upper" and "S_lower" defined?

Both were defined

- Pg8, L254-258:  A scheme would be useful.

See Figure S2a in Supplementary material

- Pg8, L266-270:  What about uncertainty?

The case study was removed and another approach was chosen, see Section 2.6.2

- Pg9, L286:  Can a polygon be circular?

It was decided to use the edge of the Puddy identified by the Faghmous et al. (2015)'s algorithm, this was clarified in the text: "For these Puddies, we estimated characteristics within the entire eddy volume as follows: i) for each $\delta h_{max}$ (Puddy center), a mask of the entire volume was created; ii) the area mask was calculated to identify the edge using the Faghmous algorithm as the most accurate approach (Figure S3; see sensitivity analysis in the Supplementary Material); iii) this mask was applied across all depths (at 5-meter intervals), creating an irregular cylinder between $S_{upper}$ and $S_{lower}$; iv) from the total volume enclosed by the cylinder, an average vertical profile was obtained at each time step for each variable (see  an example of vertical Puddy characterization in Figures S2b, S2c, S2d, S2e)." (Section 2.6.2).

- Pg9, L304-306:  What about a second criterion (for example, Okubo-Weiss parameter or so) to better define the eddies?

We proof used the Okubo-Weiss parameter but the Faghmous method defined the eddies better.

- Pg9, L305:  Eddies? Eddy signals? Eddy-like features? Define what you mean exactly.

The text may be confusing; thank you for pointing it out. In the text, we refer to Puddies—eddies, that have closed contours with a single $\delta h_{max}$ with a subsurface core.

- Pg11, L334:  Puddies profiles, Puddie profiles.

corrected

- Pg11, L335:  The number, The numbers.

corrected

- Table 1: Depths are usually defined as positive, in contrast with the vertical coordinate. The total area (and not just the number of pixels) can be interesting.

This was considered in the revised version of Table 1. Thank you.

- Pg12, L352: between, among.

corrected

- Table 3: Define "R".

Done

- Figure 3: You should move the labels "(a), (b),…" out of the small boxes, and also make them bigger.

The Figure 4 (before Figure 3), it was improved with larger labels.

- Pg14, L382: Corresponding, correspond.

corrected

- Pg14, L384-385: "Vertical error bars… of O2", unnecessary. It could be something like: Error bars correspond to the standard deviations.

corrected

- Pg15, L409: "()", underbars?

The mean state notation changed to $< >_{9-year}$

- Pg15, L410: "()'", primes?

The convention for the averaging process was changed to $< '>$

- Pg16, L429-431: "…the variability of… with oxygenation", dispersion of the eddy trajectories?

In the discussion in Section 4.2, we mention the factors involved in this variability

- Pg17, L436-437: "Vertical error bars… of O2", unnecessary. It could be something like: Error bars correspond to the standard deviations.

Done in line 485

- Pg18, L451: "denoted by PHYS", where? Do you mean "referred to as"?

corrected

- Pg19, L508:  positive vorticity, anticyclonic?

This is the Southern Hemisphere.

The study case was removed and changed to other analyses. Results showed in Section 3.4

- Pg 21, L544:  puddy, Puddy.

corrected

- Pg21, L550:  positive vorticity, anticyclonic?

In the Southern Hemisphere, the anticyclonic eddies have positive vorticity.

The study case was removed and changed to other analyses. Results showed in Section 3.4

- Figure 7:  Labels of the color bars are too small.

The previous Figure 7 was removed and replaced with another one related to the analyses presented in Section 3.4. This change was made due to a modification in the methodology, considering the suggestion of another reviewer.

- Figure 8:  Add numbers in the horizontal axes at the two upper rows. Background grids in the plots would be useful.

Similar to figure 7, the previous Figure 8 was removed and replaced with another one related to the analyses presented in Section 3.4. This change was made due to a modification in the methodology, considering another reviewer's suggestion.

- Pg23, L574:  puddies, Puddies.

Corrected

- Pg24, L589:  Delete "on".

Done

- Pg25, L596:  Insert comma after "waters".

Done

- Pg26, L665:  "within eddies", surface eddies?

This was clarified in the text: " within mode water eddies "

 - Pg27, L681:  due additional, due to additional.

Done -

Pg27, L682:  by-products, byproducts.

This correction was made because the nitrite and nitrous oxide in denitrification are not byproducts; they are intermediate products. In nitrification, however, they are byproducts.

In the text, they appear as: "also generate intermediate products such as $N_2O$ and $NO_2^-$"

- Pg27, L697:  Mention the resolution.

We prefer not to do it in this section.

- Figures A7 and A8:  The points of the results are too small, use different colors between each other for more clarity. Avoid overlapping of the points' labels.

We changed the reference letters for clarity, "O" for the observations and "M" for the model.

---

## Author Response (AR2)

Submitted on 26 May 2025
Anonymous referee #4

The authors characterize Poleward undercurrent eddies (PUDDIES), a subsurface eddy in the South-East Pacific Ocean, and how their biogeochemistry evolves. A particular focus is put on the dissolved oxygen as these eddies are hypothesized to influence the oxygen minimum zone in the region. To do this, they use a 9-year output of a coupled physical-biogeochemical model (ROMS+BioEBUS), configured at a horizontal resolution of 1/12 degree and a temporal resolution of 3 days. They found that PUDDIES have hypoxic cores in the open ocean, compared to suboxic cores close to the coast, and this is due to ventilation during their propagation. Ventilation occurred through lateral and vertical advection, and vertical mixing supply O2 to a lesser degree.

I have only seen the revised version of this manuscript and found the topic to be of interest, with the manuscript well written and clearly defined methods and choices (after revision from 1st review process). Indeed, complex biogeochemical models are computationally expensive, but are important to better resolve processes such as mixing processes which can influence the SMS dynamics.

We appreciate the time you took to read and review our work, as well as your helpful comments to improve the manuscript.

Specific comments:

• The comments raised by Reviewer 2 about the structure of the methods, as well as the methods themselves have been appropriately addressed by the authors.
• Figure 2 is really small with many subplots, hence making it really hard to read. I recommend extending it vertically, and reducing the space between the plots. Changing the colorbar to something else than jet will also be helpful.

Following your suggestion, Figure 2 was improved for better clarity of content, and the color map was changed to a more suitable range.

• Line 374: relative to instead of "concerning"

The sentence was modified  as follows: "....and estimated anomalies of the profiles relative to the general mean profile of the corresponding box".

• Line 380: I'm not sure of the word 'evidence' here. Consider changing it.

The line was modified as follows: "However, $NO_3^-$ and $N_2O$ exhibit maximum levels in the central region"

• Line 532: the case that <AOU'> is greater than…. <delta NO3> is greater. Please check these lines.

The paragraph was reviewed and the arguments are correct, therefore it was not modified.

• Line 619: Consider changing "it finishes completely mixed" to "which becomes completely mixed"

Done